# TimeStep Master: Asymmetrical Mixture of Timestep LoRA Experts for Versatile and Efficient Diffusion Models in Vision

## Abstract

Diffusion models have driven the advancement of vision generation over the past years. However, it is often difficult to apply these large models in downstream tasks, due to massive fine-tuning cost. Recently, Low-Rank Adaptation (LoRA) has been applied for efficient tuning of diffusion models. Unfortunately, the capabilities of LoRA-tuned diffusion models are limited, since the same LoRA is used for different timesteps of the diffusion process. To tackle this problem, we introduce a general and concise TimeStep Master (TSM) paradigm with two key fine-tuning stages. In the fostering stage (1-stage), we apply different LoRAs to fine-tune the diffusion model at different timestep intervals. This results in different TimeStep LoRA experts that can effectively capture different noise levels. In the assembling stage (2-stage), we design a novel asymmetrical mixture of TimeStep LoRA experts, via core-context collaboration of experts at multi-scale intervals. For each timestep, we leverage TimeStep LoRA expert within the smallest interval as the core expert without gating, and use experts within the bigger intervals as the context experts with time-dependent gating. Consequently, our TSM can effectively model the noise level via the expert in the finest interval, and adaptively integrate contexts from the experts of other scales, boosting the versatility of diffusion models. To show the effectiveness of our TSM paradigm, we conduct extensive experiments on three typical and popular LoRA-related tasks of diffusion models, including domain adaptation, post-pretraining, and model distillation. Our TSM achieves the state-of-the-art results on all these tasks, throughout various model structures (UNet, DiT and MM-DiT) and visual data modalities (Image and Video), showing its remarkable generalization capacity.

## 1 Introduction

Diffusion models have shown remarkable success in vision generation (Rombach et al., 2022b; Podell et al., 2023; Singer et al., 2022; Ho et al., 2022a; Chen et al., 2024d). Especially with the guidance of scaling law, they demonstrate the great power in generating images and videos from user prompts (Esser et al., 2024b; Liu et al., 2024a;c; Bao et al., 2024) owing to billions of model parameters. However, it is often difficult to deploy these diffusion models efficiently in various downstream tasks, since fine-tuning such huge models is resource-consuming. To fill this gap, Low-Rank Adaptation (LoRA) (Hu et al., 2021), initially developed in NLP (Chowdhary & Chowdhary, 2020), has been applied to diffusion models for rapid adaptation and efficient visual generation (Luo et al., 2023a; Li et al., 2024; Peng et al., 2024; Yin et al., 2024b).

However, we observe that the generative capability of LoRA-tuned diffusion models is limited. For illustration, we take the well-known PixArt-$\alpha$ (Chen et al., 2024d) as an example, which is pre-trained on SAM-LLaVA-Captions10M (Chen et al., 2024d) for image generation. As shown in Fig. 1, we perform LoRA on two typical fine-tuning settings. On one hand, we fine-tune this model with LoRA on new image data (*e.g.*, T2I-CompBench (Huang et al., 2023)). In this setting of downstream adaptation, the LoRA-tuned model makes similar errors as the pre-trained model, *i.e.*, they both fail to fit the target data distribution. On the other hand, we fine-tune this model with LoRA on the pretraining image data. In this setting of post-pretraining, LoRA-tuned model results in prompt misalignment, which deteriorates the generative capacity of the pre-trained model. Based on

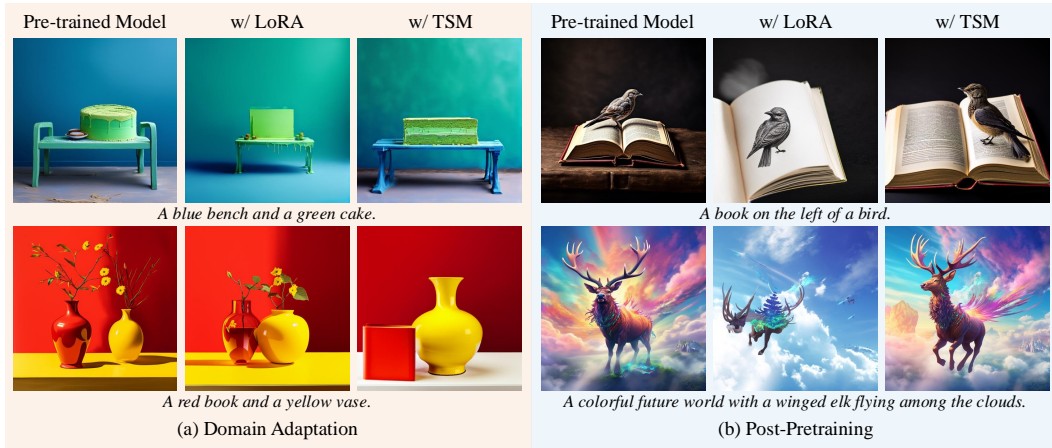

| Pre-trained Model | w/ LoRA | w/ TSM | Pre-trained Model | w/ LoRA | w/ TSM |

*A blue bench and a green cake.* — *A book on the left of a bird.*

*A red book and a yellow vase.* — *A colorful future world with a winged elk flying among the clouds.*

(a) Domain Adaptation — (b) Post-Pretraining

Figure 1: **Comparison on Image Modality.** (a) The pre-trained model and LoRA-tuned model incorrectly generate green bench and red vase, while TSM corrects these errors. (b) LoRA-tuned model generates degraded images, while TSM benefits visual quality and text alignment.

these observations, there is a natural question: *why does such deterioration appear in LoRA-tuned diffusion models*? We believe this is due to the distinct learning manner of diffusion models, *i.e.*, diffusion models process inputs with varying noise levels differently at each timestep (Balaji et al., 2022; Xue et al., 2024; Hang et al., 2023). In the vanilla LoRA setting, only ONE LoRA is applied for fine-tuning diffusion models at DIFFERENT timesteps. Thus, in the downstream adaptation case, it fails to fit the new target data just like the pre-trained model. In the post-pretraining case, such an inconsistent manner would reduce the capability of diffusion models to tackle different noise levels, especially with very limited parameters in LoRA (more evidence provided in Tab. 1 and 2).

To alleviate this problem, we propose a general and concise TimeStep Master (TSM) paradigm, with a novel asymmetrical mixture of TimeStep LoRA experts. Specifically, our TSM contains two distinct stages of fostering and assembling TimeStep LoRA experts, boosting the versatility and efficiency of tuning diffusion models in vision. In the fostering stage, we divide the training procedure into several timestep intervals. For different intervals, we introduce different LoRA modules for fine-tuning the diffusion model, leading to different TimeStep LoRA experts. This can effectively enhance the diffusion model to fit the data distribution under different noise levels. In the assembling stage, we combine the TimeStep LoRA experts of multi-scale intervals to further boost performance. Specifically, we introduce a novel asymmetrical mixture of TimeStep LoRA experts, for core-context expert collaboration. For each timestep, we leverage TimeStep LoRA expert within the smallest interval as the core expert without gating, and use experts within the bigger intervals of other scales as the context experts with time-dependent gating. In this case, our TSM can effectively learn the noise level via the expert in the finest interval, as well as adaptively integrate contexts from the experts of other scales, boosting the versatility and generalization capacity of diffusion model.

To show the effectiveness of our TSM paradigm, we conduct extensive experiments on three typical and popular LoRA-related tasks of diffusion models, including domain adaptation, post-pretraining, and model distillation. Our TSM achieves the state-of-the-art results on all these tasks, throughout various model structures (UNet (Ronneberger et al., 2015), DiT (Peebles & Xie, 2023), MM-DiT (Esser et al., 2024a)) and visual data modalities (Image, Video), showing its remarkable generalization capacity. For the above three tasks, TSM achieves the best performance on T2I-CompBench, efficiently improves model performance after post-pretraining using only public datasets, and reaches the FID of 9.90 on COCO2014 with a very low resource consumption of 3.7 A100 days.

## 2  RELATED WORK

**Diffusion models for visual synthesis.** Recently, diffusion models (DMs) have swept across the realm of visual generation and have become the new state-of-the-art generative models for text-to-image (Podell et al., 2023; Nichol et al., 2021; Li et al., 2023; Saharia et al., 2022; Chen et al., 2024d;b;c; Xue et al., 2024) and text-to-video (Ho et al., 2022b; Blattmann et al., 2023; Khachatryan et al., 2023; Luo et al., 2023b; Wang et al., 2023; Singer et al., 2022; Chen et al., 2023a;

Zhuang et al., 2024). Stable Diffusion 1.5 (SD1.5) (Rombach et al., 2022b) operates in the latent space and can generate high-resolution images. The PixArt series (Chen et al., 2024d;b;c) provide more accessibility in high-quality image generation by introducing efficient training and inference strategies. SD3 (Esser et al., 2024b) demonstrates even more astonishing generation results with the MM-DiT architecture and scaled-up parameters. VideoCrafter2 (VC2) (Chen et al., 2024a) discovers the spatial-temporal relationships of the video diffusion model and further proposes an effective training paradigm for high-quality video generation. However, the increasing number of parameters of the DMs also makes it difficult to directly transfer its powerful capabilities to other domains.

**Efficient tuning of diffusion models.** To reduce the cost of full fine-tuning DMs in downstream tasks and retaining the generalization ability, LoRA (Hu et al., 2021) is widely applied on DMs to efficiently train low-rank matrices (Zhang et al., 2023; Ye et al., 2023; Xie et al., 2023; Mou et al., 2024; Lin et al., 2024a; Xing et al., 2024; Ran et al., 2024; Gu et al., 2024; Lyu et al., 2024; Huang et al., 2023). GORS (Huang et al., 2023) applys LoRA to finetune the DMs to the target domain. DMD (Yin et al., 2024b) supports the use of LoRA in model distillation for fast inference. ControlNeXt (Peng et al., 2024) employed LoRA for efficient and enhanced controllable generation. T2V-Turbo (Li et al., 2024) injected LoRA to video diffusion model (Chen et al., 2024a) and optimized with mixed rewards, achieving inference acceleration and quality improvement. But as discussed earlier, the generation capabilities of LoRA-tuned DMs are limited. We tackle this with our TSM, which assigns TimeStep LoRA experts to learn the distribution within diverse noise levels, and assemble these experts for further information aggregation. Using TSM, the generative performance of pre-trained diffusion models is significantly enhanced at a low fine-tuning cost.

## 3 METHOD

In this section, we introduce our TimeStep Master (TSM) paradigm in detail. First, we briefly review the diffusion model and LoRA as preliminaries. Then, we explain two key fine-tuning stages in TSM, *i.e.*, expert fostering and assembling, in order to build an asymmetrical mixture of TimeStep LoRA experts for efficient and versatile enhancement of the diffusion model.

**Diffusion Model.** The diffusion model is designed to learn a data distribution by gradually denoising a normally-distributed variable (Song et al., 2021; Ho et al., 2020). It has been widely used for image/video generation (Rombach et al., 2022b; Podell et al., 2023; Singer et al., 2022; Ho et al., 2022a; Chen et al., 2024d; Zhuang et al., 2024; Chen et al., 2024e). In the forward diffusion process, one should add Gaussian noise $\epsilon \sim \mathcal{N}(0, I)$ on the input $x_0$, in order to generate the noisy input $x_t$ at each timestep, $x_t = \sqrt{\overline{\alpha}_t} x_0 + \sqrt{1 - \overline{\alpha}_t} \epsilon$, where $t = 1, 2, \cdots, T$, and $T$ is the total number of timesteps in the forward process. $\overline{\alpha}_t$ is a parameter related to $t$. When $t$ approaches $T$, $\overline{\alpha}_t$ approaches 0. The training goal is to minimize the loss function for denoising,

$$\mathcal{L} = \mathbf{E}_{x_0, c, \epsilon, t} \left[ \| \epsilon - \epsilon_\Theta(x_t, t, c) \|_2^2 \right], \quad t \in [1, T], \tag{1}$$

where $\epsilon_\Theta$ is the output of neural network with model parameters $\Theta$, and $c$ indicates the additional condition, *e.g.*, text input. To achieve superior performance, the diffusion model is often designed with a large number of network parameters that are pre-trained on large-scale web data. Apparently, it is computationally expensive to fine-tune such a big model for specific downstream tasks.

**Low-Rank Adaptation (LoRA).** To alleviate the above difficulty, LoRA (Hu et al., 2021) has been recently applied for rapid fine-tuning diffusion models on target data (Ruiz et al., 2023; Huang et al., 2023). Specifically, LoRA introduces low-rank decomposition of an extra matrix,

$$\Theta + \Delta\Theta = \Theta + BA, \tag{2}$$

where $\Theta \in R^{d \times k}$ is the pretrained parameter matrix of diffusion model. $\Delta\Theta \in R^{d \times k}$ is the extra parameter matrix that is decomposed as the multiplication of two low-rank matrices $A \in R^{r \times k}$ and $B \in R^{d \times r}$, where $r \ll d, k$. To achieve parameter-efficient fine-tuning, one can simply freeze the pre-trained parameter $\Theta$, while only learning the low-rank matrices $A$ and $B$ on target data for computation cost reduction. However, the generation capabilities of these vanilla LoRA-tuned diffusion models are limited. The main reason is that, diffusion model exhibits different processing modes for the noisy inputs at different timesteps (Balaji et al., 2022; Hang et al., 2023). Alternatively, LoRA applies the same low-rank matrices $A$ and $B$ for different timesteps. Such inconsistency would reduce the capacity of diffusion model to tackle different noise levels, especially with a very limited number of learnable parameters in $A$ and $B$. To address this problem, we propose a TimeStep Master (TSM) paradigm with two important stages as follows.

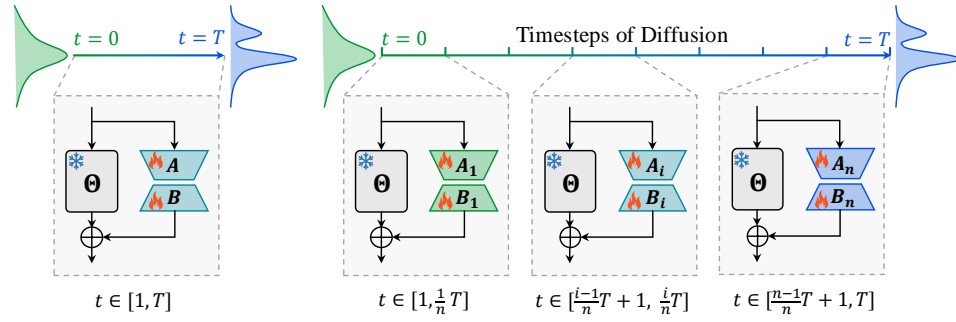

$t \in [1, T]$    $t \in [1, \frac{1}{n}T]$    $t \in [\frac{i-1}{n}T + 1, \frac{i}{n}T]$    $t \in [\frac{n-1}{n}T + 1, T]$

**(a) Vanilla LoRA**    **(b) TimeStep LoRA Experts of $n$ intervals**

Figure 2: **Fostering Stage: TimeStep LoRA Expert Construction.** We divide all $T$ timesteps into $n$ intervals and fine-tune the diffusion model with individual LoRA module for each interval.

## 3.1 FOSTERING STAGE: TIMESTEP LORA EXPERT CONSTRUCTION

To learn different modes of the noisy inputs, we propose to introduce different LoRAs for different timesteps. Specifically, we uniformly divide the timesteps of $T$ into $n$ intervals. For the $i$-th interval, we introduce an individual LoRA,

$$\Theta + \Delta\Theta_i = \Theta + B_i A_i \tag{3}$$

where $A_i \in R^{r \times k}$ and $B_i \in R^{d \times r}$ refer to low-rank matrices in the $i$-th interval. We optimize $A_i$ and $B_i$ by fine-tuning the diffusion model on the noisy inputs within the $i$-th interval,

$$\mathcal{L} = \mathbf{E}_{x_0, c, \epsilon, t} \left[ \| \epsilon - \epsilon_{\Theta, A_i, B_i} (x_t, t, c) \|_2^2 \right], \quad t \in [\frac{i-1}{n} \cdot T + 1, \ \frac{i}{n} \cdot T]. \tag{4}$$

We dub the fine-tuned diffusion model as a TimeStep LoRA expert at interval $i$. Hence, we can obtain $n$ TimeStep LoRA experts for $n$ intervals of timesteps. During inference, we first sample $x_T$ from Gaussian noise $x_T \sim \mathcal{N}(0, I)$, and then use these TimeStep LoRA experts to iteratively denoise $x_T$, i.e., when the timestep $t$ iterates to one certain interval, we use the corresponding TimeStep LoRA expert of this interval to estimate the noise of $x_t$, where $t = T, ..., 1$.

It is worth mentioning that, there are two extreme cases with $n = 1$ and $n = T$. When $n = 1$, it refers to the vanilla LoRA setting that is limited to capture different noise levels at different timesteps. When $n = T$, it refers to the setting where there is a LoRA expert for each timestep. Apparently, this setting makes no sense since the noise levels are similar among the adjacent timesteps. Hence, it is unnecessary to equip a LoRA for each timestep. Especially $T$ is often large in the diffusion model, such an extreme setting introduces too many LoRA parameters to learn. Consequently, we propose to divide $T$ in different numbers of intervals, i.e., $n = n_1, n_2, \cdots, n_m$. In this case, for each timestep $t$, there are $m$ TimeStep LoRA experts. In the following, we introduce a novel asymmetrical mixture of these TimeStep LoRA experts, which can effectively and adaptively make them collaborate to further boost diffusion models via multi-scale noise modeling.

## 3.2 ASSEMBLING STAGE: ASYMMETRICAL MIXTURE OF TIMESTEP LORA EXPERTS

Via the multi-scale design of interval division above, one can obtain $m$ TimeStep LoRA experts for each timestep $t$. Hence, the next question is how to assemble their power to model the noise level of this step. Naively, one can leverage the standard Mixture of Experts (MoE) (Riquelme et al., 2021; Chen et al., 2023b) without distinguishing the role of experts. But this is not the case for TimeStep LoRA experts. Apparently, for each timestep, the TimeStep LoRA expert within the smallest interval plays the core role in modeling the noise level of this step with fine granularity. When the interval is bigger, the granularity of noise modeling is getting bigger, i.e., the TimeStep LoRA experts within bigger intervals are getting more insensitive to noise levels.

Based on this analysis, we introduce a novel and concise asymmetrical mixture of TimeStep LoRA experts for core-context expert collaboration. Specifically, for each timestep $t$, we leverage TimeStep LoRA expert within the smallest interval as the core expert without gating, and use the rest $(m-1)$ experts as the context ones with gating,

$$\Theta + \Delta\Theta_{i_1} + \mathcal{G}(z_t, t) \odot [\Delta\Theta_{i_2}, ...., \Delta\Theta_{i_m}] = \Theta + B_{i_1} A_{i_1} + \sum\nolimits_{j=2}^{m} \mathcal{G}_j \odot B_{i_j} A_{i_j}, \tag{5}$$

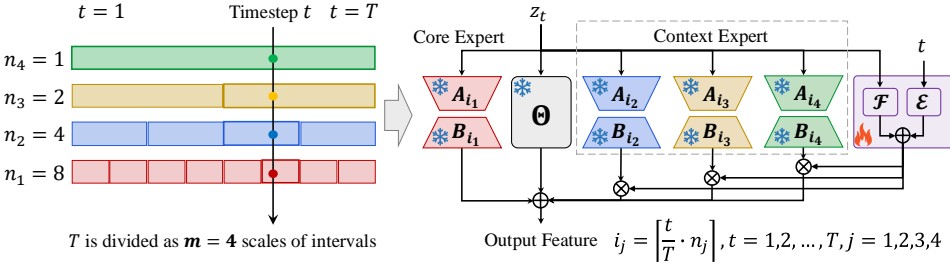

Figure 3: **Assembling Stage: Asymmetrical Mixture of TimeStep LoRA Experts.** We divide $T$ into 4 intervals, namely $n_1{=}8$, $n_2{=}4$, $n_3{=}2$, $n_4{=}1$. The TimeStep LoRA expert within the smallest-scale interval plays the core role to model the noise level of $t$ with fine granularity. The core expert (red) is without gating; the context experts (blue, yellow and green) are with gating. The router is timestep-dependent, which adaptively weights the importance of context experts at $t$.

Note that, we design the router of gating $\mathcal{G}(z_t, t) \in R^{m-1}$ to be timestep dependent, in order to adaptively weight contexts of the rest $(m-1)$ experts according to the timestep. Specifically, we make $\mathcal{G}(z_t, t)$ as a transformation of the timestep $t$ and the input feature $z_t \in R^{k \times l}$ of this step. For simplicity, we design it as the sum over a FC layer of $z_t$ and an embedding layer of $t$,

$$\mathcal{G}(z_t, t) = [\mathcal{G}_2, ..., \mathcal{G}_m] = \mathcal{F}(z_t) + \mathcal{E}(t), \tag{6}$$

where the embedding layer refers to a learnable matrix with a size of $T \times (m-1)$, and $\mathcal{E}(t)$ means that we extract the parameters in the $t$-th row as the embedding of timestep $t$. Finally, we minimize the diffusion loss function over this asymmetrical mixture of TimeStep LoRA experts,

$$\mathcal{L} = \mathbf{E}_{x_0, c, \epsilon, t} \left[ \| \epsilon - \epsilon_{\Theta, \{A_{i_j}, B_{i_j}\}_{j=1}^m, \mathcal{G}} (x_t, t, c) \|_2^2 \right], \quad i_j = \lceil \frac{t}{T} \cdot n_j \rceil, \tag{7}$$

where the timestep $t$ simultaneously belongs to intervals of $m$ scales, *i.e.*, $t = 1, 2, \cdots, T$, and $j = 1, ..., m$. Note that, the TimeStep LoRA experts have been trained in the fostering stage. Hence, we freeze them and only learn the parameters of router $\mathcal{G}(z_t, t)$ in the assembling stage. Via such a distinct paradigm, our TSM can further boost diffusion to master noise modeling via TimeStep expert collaboration, as well as inherit the efficiency of LoRA for rapid adaption.

## 4 EXPERIMENTS

We apply Timestep Master (TSM) to three typical fine-tuning tasks of diffusion model in visual generation: **domain adaptation**, **post-pretraining**, and **model distillation**. Extensive results demonstrate TSM achieves the state-of-the-art performance on all these tasks, throughout different model structures and modalities. We also make detailed ablation and visualization to show its effectiveness.

### 4.1 DOMAIN ADAPTATION

**Problem Definition and Dataset**. Domain adaptation (Farahani et al., 2021) refers to the task of adapting a model trained on a source domain to perform well on a different but related target domain. The goal is to fit the target domain distribution while preserving the strong generalization ability of the pre-trained model. We conduct domain adaptation experiments on T2I-CompBench (Huang et al., 2023), an open-world text-to-image generation benchmark which contains six domains. Each domain includes domain-specific training and testing prompts (700:300) and employs specialized models to evaluate generated test images and we convert all scores into percentile for ease of reading.

**Implementation Details.** Following (Huang et al., 2023), we generate 90 distinct 512x512 resolution images per training prompt for adaptation. We conduct both vanilla LoRA (Hu et al., 2021) and TSM experiments based on the pre-trained models of SD1.5 (Rombach et al., 2022a), PixArt-$\alpha$ (Chen et al., 2024d) and Stable Diffusion 3 (SD3) (Esser et al., 2024b). For SD3, in vanilla LoRA and TSM fostering stage (1-stage), we employ LoRA on the *to_q*, *to_k*, *to_v* and *to_out.0* modules of the MM-DiT and *q_proj*, *k_proj*, *v_proj* and *out_proj* modules of two CLIP text encoders (Radford et al., 2021a; Cherti et al., 2023). We set LoRA $r, \alpha{=}4$, and employ zero initialization for all matrix $B$. At TSM assembling stage (2-stage), we add router to the module which is equipped with LoRA

| Method | Color↑ | Shape↑ | Texture↑ | Spatial↑ | Non-Spatial↑ | Complex↑ |
|---|---|---|---|---|---|---|
| SD1.4 (Rombach et al., 2022b) | 37.65 | 35.76 | 41.56 | 12.46 | 30.79 | 30.80 |
| SD1.5 (Rombach et al., 2022b) | 36.97 | 36.27 | 41.25 | 11.04 | 31.05 | 30.79 |
| SD2 (Rombach et al., 2022b) | 50.65 | 42.21 | 49.22 | 13.42 | 31.27 | 33.86 |
| SD2 + Composable (Liu et al., 2022) | 40.63 | 32.99 | 36.45 | 8.00 | 29.80 | 28.98 |
| SD2 + Structured (Yu et al., 2023) | 49.90 | 42.18 | 49.00 | 13.86 | 31.11 | 33.55 |
| SD2 + Attn Exct (Wang et al., 2024) | 64.00 | 45.17 | 59.63 | 14.55 | 31.09 | 34.01 |
| SD2 + GORS unbaised (Huang et al., 2023) | 64.14 | 45.46 | 60.25 | 17.25 | 31.58 | 34.70 |
| SD2 + GORS (Huang et al., 2023) | 66.03 | 47.85 | 62.87 | 18.15 | 31.93 | 33.28 |
| SDXL (Podell et al., 2023) | 58.79 | 46.87 | 52.99 | 21.33 | 31.19 | 32.37 |
| PixArt-$\alpha$ (Chen et al., 2024d) | 41.70 | 37.96 | 45.27 | 19.89 | 30.74 | 33.43 |
| PixArt-$\alpha$-ft (Chen et al., 2024d) | 66.90 | 49.27 | 64.77 | 20.64 | **31.97** | 34.33 |
| DALLE3 (Betker et al., 2023) | 77.85 | 62.05 | 70.36 | 28.65 | 30.03 | 37.73 |
| SD3 (Esser et al., 2024b) | 80.33 | 58.49 | 74.27 | 26.44 | 31.43 | 38.62 |
| SD1.5 + Vanilla LoRA (Hu et al., 2021) | 51.70 | 44.76 | 52.68 | 15.45 | 31.69 | 32.83 |
| PixArt-$\alpha$ + Vanilla LoRA (Hu et al., 2021) | 46.53 | 43.75 | 53.37 | 23.08 | 30.97 | 34.75 |
| SD3 + Vanilla LoRA (Hu et al., 2021) | 82.41 | 62.32 | 77.27 | 31.87 | 31.72 | 38.41 |
| SD1.5 + TSM (Ours) | 57.12 | 46.65 | 58.16 | 18.80 | 31.83 | 32.94 |
| PixArt-$\alpha$ + TSM (Ours) | 54.66 | 44.47 | 57.12 | 25.41 | 31.05 | 34.85 |
| SD3 + TSM (Ours) | **83.45** | **63.16** | **78.18** | **34.50** | 31.81 | **38.71** |

Table 1: **Domain Adaptation on T2I-CompBench.** Our TSM demonstrates the best performance in terms of color, shape, texture, spatial and complex, outperforming SOTA methods.

| Image Modality Method | Color↑ | Shape↑ | Texture↑ | Spatial↑ | Non-Spatial↑ | Complex↑ |
|---|---|---|---|---|---|---|
| PixArt-$\alpha$ (Chen et al., 2024d) | 41.70 | 37.96 | 45.27 | 19.89 | 30.74 | **33.43** |
| + LoRA (Hu et al., 2021) | 43.47 ↑ 1.77 | 34.74 ↓ 3.22 | 41.57 ↓ 3.70 | 15.37 ↓ 4.52 | 30.74 | 30.43 ↓ 3.00 |
| + TSM (Ours) | **48.86** ↑ 7.16 | **37.97** ↑ 0.01 | **47.31** ↑ 2.04 | **21.55** ↑ 1.66 | **31.13** ↑ 0.39 | 32.96 ↓ 0.47 |

| Video Modality Method | IS↑ | Action↑ | Amplitude↑ | BLIP-BLEU↑ | Color↑ | Count↑ |
|---|---|---|---|---|---|---|
| VC2 (Chen et al., 2024a) | 16.76 | 77.76 | 44.0 | 23.02 | 46.74 | 53.77 |
| + LoRA (Hu et al., 2021) | 15.06 ↓ 1.70 | 73.85 ↓ 3.91 | 46.0 ↑ 2.0 | 21.89 ↓ 1.13 | 41.30 ↓ 5.44 | 27.89 ↓ 25.88 |
| + TSM (Ours) | **18.08** ↑ 1.32 | **80.77** ↑ 3.01 | **54.0** ↑ 10.0 | **24.26** ↑ 1.24 | **60.87** ↑ 14.13 | **60.38** ↑ 6.61 |

Table 2: **Image and Video Modality Post-Pretraining on T2I-CompBench and EvalCrafter.** Our TSM continues to improve model performance compared to vanilla LoRA.

and set TimeStep experts $n_1=8$, $n_2=1$. We train 4K steps for vanilla LoRA and two stages of TSM. The global batch size is 64. We use the AdamW optimizer with $\beta_1=0.9$, $\beta_2=0.999$. For MM-DiT, the learning rate is set to 1e-5 and the weight decay to 1e-4. For text encoder, the learning rate is set to 5e-6 and the weight decay to 1e-3. The settings of SD1.5 and PixArt-$\alpha$ are in Sec. 4.4, 4.2.

As shown in Tab. 1, TSM achieves state-of-the-art results on T2I-CompBench and is far ahead in domains of color, shape, texture, and spatial. For complex domain, which contains more complex prompts and metrics than others, the performance of the model deteriorates after employing vanilla LoRA for domain adaptation. However, TSM can still improve the model performance.

### 4.2 POST-PRETRAINING

**Problem Definition and Dataset**. Post-pretraining (Luo et al., 2022) refers to the task of continuing to train a pre-trained model on a general dataset. The goal is to further improve the general performance of the model. We conduct experiments on post-pretraining tasks in both image and video modalities. For image modality, we evaluate our post-trained model on T2I-CompBench (Huang et al., 2023) as in Sec. 4.1. For video modality, we use EvalCrafter (Liu et al., 2024b), a public benchmark for text-to-video generation using 700 diverse prompts. Specifically, we adopt Inception Score (IS) for video quality assessment. For motion quality, we consider Action Recognition (Action) and Amplitude Classification Score (Amplitude). We evaluate text-video alignment with Text-Text Consistency (BLIP-BLEU) and Object and Attributes Consistency (Color and Count).

**Implementation Details**. For image modality, we conduct both vanilla LoRA (Hu et al., 2021) and TSM experiments based on the pre-trained model PixArt-$\alpha$ Chen et al. (2024d) and the training dataset SAM-LLaVA-Captions 10M (Chen et al., 2024d). In vanilla LoRA and TSM 1-stage, we employ LoRA on the *to_q*, *to_k*, *to_v* and *to_out.0* modules of the DiT (Peebles & Xie, 2022) and *q,v* modules of T5 text encoder (Raffel et al., 2020). For model and training settings, we adopt the same LoRA and router strategies as SD3 in Sec. 4.1 for vanilla LoRA and TSM. The learning rate is 2e-5 and the weight decay is 1e-2 for both DiT and text encoder. For video modality, we conduct experiments based on the pre-trained VideoCrafter2 (Chen et al., 2024a) and use a 70k subset of OpenVid-1M (Nan et al., 2024) for post-pretraining. In vanilla LoRA and TSM 1-stage, we inject

| Family | Method | Resolution↑ | $N_{\text{params}}$↓ | Training Cost↓ | FID↓ |
|---|---|---|---|---|---|
| **Unaccelerated Diffusion** | DALL-E (Ramesh et al., 2021) | 256 | 12.0B | 2048 V100 × 3.4M steps | 27.5 |
| | DALL-E 2 (Ramesh et al., 2022) | 256 | 6.5B | 41667 A100 days | 10.39 |
| | Make-A-Scene (Gafni et al., 2022) | 256 | 4.0B | - | 11.84 |
| | GLIDE (Nichol et al., 2021) | 256 | 5.0B | - | 12.24 |
| | LDM (Rombach et al., 2022b) | 256 | 1.45B | - | 12.63 |
| | Imagen (Saharia et al., 2022) | 256 | 7.9B | 4755 TPUv4 days | 7.27 |
| | eDiff-I (Balaji et al., 2022) | 256 | 9.1B | 256 A100 × 600K steps | 6.95 |
| | SD1.5 (50 step, cfg=3, ODE) | 512 | 860M | 6250 A100 days | 8.59 |
| | SD1.5 (200 step, cfg=2, SDE) | 512 | 860M | 6250 A100 days | 7.21 |
| **Accelerated Diffusion** | DPM++ (Lu et al., 2022) | 512 | - | - | 22.36 |
| | UniPC (4 step) (Zhao et al., 2024) | 512 | - | - | 19.57 |
| | LCM-LoRA (4 step) (Luo et al., 2023a) | 512 | 67M | 1.3 A100 days | 23.62 |
| | InstaFlow-0.9B (Liu et al., 2023) | 512 | 0.9B | 199 A100 days | 13.10 |
| | SwiftBrush (Nguyen & Tran, 2024) | 512 | 860M | 4.1 A100 days | 16.67 |
| | HiPA (Zhang & Hooi, 2023) | 512 | 3.3M | 3.8 A100 days | 13.91 |
| | UFOGen (Xu et al., 2024b) | 512 | 860M | - | 12.78 |
| | SLAM (4 step) (Xu et al., 2024a) | 512 | 860M | 6 A100 days | 10.06 |
| | DMD (Yin et al., 2024b) | 512 | 860M | 108 A100 days | 11.49 |
| | DMD2 (Yin et al., 2024a) | 512 | 860M | 70 A100 days | 8.35 |
| | DMD2 + LoRA (Hu et al., 2021) | 512 | 67M | 3.6 A100 days | 14.58 |
| | DMD2 + TSM (Ours) | 512 | 68M | 3.7 A100 days | **9.90** |

Table 3: **Model Distillation on 30K prompts from COCO2014.** Our TSM achieves competitive FID compared to SOTA models while lowering the training cost significantly. Rows marked in gray demonstrate the superiority of our TSM over the vanilla LoRA based on DMD2.

LoRA on the *k, v* modules in both spatial and temporal layers of the 3D-UNet and *out_proj* module of OpenCLIP (Cherti et al., 2023) text encoder. We set LoRA $r, \alpha=16$ and adopt $lora\_dropout=0.01$ only in the 3D-UNet. In TSM 2-stage, we add router to the module where LoRA is injected and set TimeStep experts $n_1=8, n_2=4$. We train 5K steps for vanilla LoRA and two stages of TSM. The global batch size is 32. We use the same optimizer setting as in image modality. The learning rate is 2e-4 and the weight decay is 1e-2 for both UNet and text encoder.

As shown in Tab. 2, the performance of models using vanilla LoRA for post-pretraining drops significantly. TSM continues to improve model performance without higher quality internal data.

### 4.3 MODEL DISTILLATION

**Problem Definition and Dataset**. Model distillation (Gou et al., 2021) refers to the task of training a simplified and efficient model to replicate the behavior of a complex one. Since LoRA is widely used in model distillation, we explore the capabilities of TSM in this task. We conduct experiments on 30K prompts from COCO2014 (Lin et al., 2014) validation set. Following DMD2 (Yin et al., 2024a), we generate images from these prompts and compare these images with 40,504 real images from the same validation set to calculate the Fréchet Inception Distance (FID) (Heusel et al., 2017).

**Implementation Details.** We distill a 4-step (i.e., 999, 749, 499, 249) generator from 1000 steps of SD1.5 (Rombach et al., 2022b). Following DMD2, we first train the model without a GAN loss, and then with the GAN loss on 500K real images from LAION-Aesthetic (Schuhmann et al., 2022). We employ LoRA with $r=64$, $\alpha=8$ on *to_q*, *to_k*, *to_v*, *to_out.0*, *proj_in*, *proj_out*, *ff.net.0.proj*, *ff.net.2*, *conv1*, *conv2*, *conv_shortcut*, *downsamplers.0.conv*, *upsamplers.0.conv* and *time_emb_proj* modules of UNet. In vanilla LoRA, we train for 40K steps without GAN loss and 5K steps with it. In TSM 1-stage, we train the experts at 999 and 749 timesteps for 20K steps without GAN loss and 5K steps with it. At 499 and 249 timesteps, we reduce training without GAN loss to 5K steps and increase training with real image guidance to 20K and 40K steps respectively. In TSM 2-stage, we train the router and freeze other modules with $n_1=4$, $n_2=1$ TimeStep experts. We only train it for 2K steps with GAN loss, due to the little $N_{\text{params}}$ (<1M). The batch size is 32 without GAN loss and 16 with it (4 times for vanilla LoRA). Other settings are consistent with DMD2.

Tab. 3 shows the SOTA comparison on model distillation, where $N_{\text{params}}$ refers to the trainable parameters and *Training Cost* is calculated based on a single A100 GPU. Notably, our TSM far outperforms LoRA (FID 9.90 *vs.* 14.58) with an increase of less than 1M trainable parameters and 0.1 A100 days gain of training cost. Although we could not achieve the lowest FID due to our limited training resources, we obtain a competitive result while significantly reducing the training cost. This demonstrates the effectiveness and efficiency of our TSM in model distillation.

| Model | FT Method | Color↑ | Shape↑ | Texture↑ | Spatial↑ | Non-Spatial↑ | Complex↑ |
|---|---|---|---|---|---|---|---|
| SD1.5 UNet | Vanilla LoRA | 51.57 | 44.76 | 52.68 | 15.45 | 31.69 | 32.83 |
| | TSM 1-stage | 56.48↑4.91 | 45.91↑1.15 | 57.08↑5.12 | 18.01↑2.56 | 31.77↑0.08 | 32.79↓0.04 |
| | TSM 2-stage | 57.12↑5.55 | 46.65↑1.89 | 58.16↑5.48 | 18.80↑3.35 | 31.83↑0.14 | 32.94↑0.11 |
| PixArt-α DiT | Vanilla LoRA | 46.53 | 43.75 | 53.37 | 23.08 | 30.97 | 34.75 |
| | TSM 1-stage | 52.84↑6.31 | 43.92↑0.17 | 54.07↑0.7 | 25.35↑2.27 | 31.03↑0.06 | 35.04↑0.29 |
| | TSM 2-stage | 54.66↑8.13 | 44.47↑0.72 | 57.12↑3.75 | 25.41↑2.33 | 31.05↑0.08 | 34.85↑0.10 |
| SD3 MM-DiT | Vanilla LoRA | 82.41 | 62.32 | 77.27 | 31.87 | 31.72 | 38.41 |
| | TSM 1-stage | 82.52↑0.11 | 62.94↑0.62 | 77.55↑0.28 | 33.08↑1.21 | 31.74↑0.02 | 38.54↑0.13 |
| | TSM 2-stage | 83.45↑1.04 | 63.16↑0.84 | 78.18↑0.91 | 34.50↑2.63 | 31.81↑0.09 | 38.71↑0.30 |

Table 4: **Domain Adaptation Ablation on T2I-CompBench.**

| IMG Model | FT Method | Color↑ | Shape↑ | Texture↑ | Spatial↑ | Non-Spatial↑ | Complex↑ |
|---|---|---|---|---|---|---|---|
| PixArt-α | Vanilla LoRA | 43.47 | 34.74 | 41.57 | 15.37 | 30.74 | 30.43 |
| | TSM 1-stage | 45.66 ↑2.19 | 37.06 ↑2.32 | 45.42 ↑3.85 | 22.32 ↑6.95 | 31.03 ↑0.29 | 32.65 ↑2.22 |
| | TSM 2-stage | 48.86 ↑5.39 | 37.97 ↑3.23 | 47.31 ↑5.74 | 16.18 ↑1.66 | 31.13 ↑0.39 | 32.96 ↑2.53 |

| VID Model | FT Method | IS↑ | Action↑ | Amplitude↑ | BLIP-BLEU↑ | Color↑ | Count↑ |
|---|---|---|---|---|---|---|---|
| VC2 | Vanilla LoRA | 15.06 | 73.85 | 46.0 | 21.89 | 41.30 | 27.89 |
| | TSM 1-stage | 16.71 ↑1.65 | 79.07 ↑5.22 | 50.0 ↑4.0 | 23.99 ↑2.10 | 56.52 ↑15.22 | 55.48 ↑27.59 |
| | TSM 2-stage | 18.08 ↑3.02 | 80.77 ↑6.92 | 54.0 ↑8.0 | 24.26 ↑2.37 | 60.87 ↑19.57 | 60.38 ↑32.49 |

Table 5: **Image and Video Post-Pretraining Ablation on T2I-CompBench and EvalCrafter.**

| Model | $n$ | $r$ | $step$ | Color↑ | Shape↑ | Texture↑ | Spatial↑ | Non-Spatial↑ | Complex↑ |
|---|---|---|---|---|---|---|---|---|---|
| SD1.5 UNet | *w/o fine-tuning* | | | 36.97 | 36.27 | 41.25 | 11.04 | 31.05 | 30.79 |
| | 1 | 4 | 4000 | 49.10 | 44.62 | 53.62 | 14.00 | 31.69 | **33.02** |
| | 1 | 32 | 4000 | 51.70 | 44.76 | 52.68 | 15.45 | 31.69 | 32.83 |
| | 1 | 4 | 32000 | 51.86 | 44.74 | 55.74 | 15.70 | 31.70 | 29.84 |
| | 2 | 4 | 4000 | 52.02 | 43.61 | 55.43 | 16.35 | 31.74 | **33.13** |
| | 2 | 4 | 16000 | 54.30 | 45.25 | **57.26** | 17.05 | **31.79** | 31.50 |
| | 4 | 4 | 4000 | 54.24 | 45.78 | 56.61 | 17.97 | 31.73 | **33.13** |
| | 4 | 4 | 8000 | **55.85** | **46.45** | **58.06** | **18.32** | **31.77** | 32.95 |
| | 8 | 4 | 4000 | **56.48** | **45.91** | 57.08 | **18.01** | **31.77** | 32.79 |
| PixArt-α DiT | *w/o fine-tuning* | | | 41.70 | 37.96 | 45.27 | 19.89 | 30.74 | 33.43 |
| | 1 | 4 | 4000 | 46.26 | 42.58 | 52.01 | 23.00 | 30.88 | 34.58 |
| | 1 | 32 | 4000 | 46.53 | 43.75 | 53.37 | 23.08 | 30.97 | 34.75 |
| | 1 | 4 | 32000 | 52.55 | 43.47 | 53.20 | 22.95 | 31.00 | 33.67 |
| | 2 | 4 | 4000 | 50.68 | 43.69 | 54.57 | 24.41 | 30.96 | 34.76 |
| | 2 | 4 | 16000 | 53.00 | **44.43** | **55.08** | 24.95 | 31.02 | 34.63 |
| | 4 | 4 | 4000 | 51.96 | 43.42 | 53.38 | 24.76 | 31.02 | **34.98** |
| | 4 | 4 | 8000 | **52.77** | 43.77 | **55.48** | **25.64** | **31.06** | 34.68 |
| | 8 | 4 | 4000 | **52.84** | **43.92** | 54.07 | **25.35** | **31.03** | **35.04** |
| SD3 MM-DiT | *w/o fine-tuning* | | | 80.33 | 58.49 | 74.27 | 26.44 | 31.43 | **38.62** |
| | 1 | 4 | 4000 | 81.28 | 61.31 | 76.65 | 31.28 | 31.70 | 38.55 |
| | 1 | 32 | 4000 | 82.41 | 62.32 | 77.27 | 31.87 | 31.72 | 38.41 |
| | 1 | 4 | 32000 | 81.82 | 62.53 | 76.81 | 32.94 | 31.73 | **38.97** |
| | 2 | 4 | 4000 | 81.74 | 61.82 | 76.68 | 32.01 | 31.73 | 38.44 |
| | 2 | 4 | 16000 | 82.60 | 62.71 | 77.80 | 32.98 | **31.79** | 38.61 |
| | 4 | 4 | 4000 | 82.24 | 62.00 | 77.11 | 32.20 | **31.79** | 38.35 |
| | 4 | 4 | 8000 | **82.76** | **62.77** | **77.57** | **33.01** | 31.75 | 38.54 |
| | 8 | 4 | 4000 | **82.52** | **62.94** | **77.55** | **33.08** | 31.74 | 38.54 |

Table 6: **TSM 1-Stage Ablation.** $n$, $r$ and $step$ represent the number, rank and fine-tuning steps of TimeStep experts. Values in **red** and **blue** represent the optimal and suboptimal respectively. When $n$=1, TSM 1-stage is equal to vanilla LoRA; when $n$>1, it significantly outperforms vanilla LoRA.

## 4.4 ABLATION STUDIES

**Overall Design.** We conduct two-stage ablation experiments on domain adaptation, post-pretraining, and model distillation. As shown in Tab. 4, in domain adaptation, our TSM significantly outperforms the vanilla LoRA on three main generative model architectures (UNet, Dit, and MM-DiT), verifying the generalization of TSM on model architecture. The model and training settings of SD1.5, PixArt-α and SD3 are same as Sec. 4.4, 4.2, 4.1 respectively. As shown in Tab. 5, in post-pretraining, TSM achieves huge improvements over vanilla LoRA on two modalities (image and video), verifying the generalization of TSM on visual modality. The experimental settings are same as Sec. 4.2. As shown in Tab. 8, in model distillation, TSM outperforms the vanilla LoRA on FID, Patch-FID (Lin et al., 2024b; Chai et al., 2022), and CLIP score (Radford et al., 2021b) on 30K prompts from COCO2014, demonstrating the generality of our TSM throughout various tasks.

**Fostering Stage**. We conduct TSM 1-stage ablation experiments for TimeStep experts' $n$, $r$, and fine-tuning $step$ on T2I-CompBench, based on SD1.5, PixArt-α, and SD3. For SD1.5, in vanilla LoRA and TSM 1-stage, we employ LoRA on the *to_q*, *to_k*, *to_v* and *to_out.0* modules of the UNet and *q_proj* and *v_proj* modules of CLIP text encoders. The learning rate is 1e-4 and other model

| Model | $n_{core}$ | $n_{context}$ | Color↑ | Shape↑ | Texture↑ | Spatial↑ | Non-Spatial↑ | Complex↑ |
|---|---|---|---|---|---|---|---|---|
| | 4 | - | 55.85 | 46.45 | 58.06 | 18.32 | 31.77 | 32.95 |
| | - | 1,4 | 56.42 | 45.77 | 56.59 | 17.17 | 31.76 | 32.66 |
| | 4 | 1 | 56.93 | 46.92 | 57.95 | 18.02 | 31.79 | 32.71 |
| | 4 | 2 | 56.84 | 46.70 | 57.70 | 17.86 | 31.75 | 32.80 |
| SD1.5 | 4 | 8 | 56.96 | 46.12 | 59.00 | 18.43 | 31.74 | 32.76 |
| UNet | 8 | - | 56.48 | 45.91 | 57.08 | 18.01 | 31.77 | 32.79 |
| | - | 1,8 | 54.56 | 45.52 | 56.30 | 17.90 | 31.78 | 33.27 |
| | 8 | 1 | 57.12 | 46.65 | 58.16 | 18.70 | 31.83 | 32.94 |
| | 8 | 2 | 56.20 | 46.58 | 58.04 | 18.17 | 31.78 | 32.91 |
| | 8 | 4 | 56.63 | 46.70 | 58.80 | 18.84 | 31.77 | 32.69 |
| | 8 | 1,2,4 | 57.59 | 46.18 | 57.69 | 17.91 | 31.82 | 32.78 |
| | 4 | - | 52.77 | 43.77 | 55.48 | 25.64 | 31.06 | 34.68 |
| | - | 1,4 | 53.24 | 43.79 | 54.70 | 25.63 | 31.06 | 35.02 |
| | 4 | 1 | 53.57 | 44.29 | 56.26 | 25.55 | 31.04 | 34.58 |
| | 4 | 2 | 53.54 | 44.02 | 56.02 | 26.17 | 31.08 | 34.41 |
| PixArt-$\alpha$ | 4 | 8 | 52.70 | 43.66 | 55.62 | 25.37 | 31.06 | 34.68 |
| DiT | 8 | - | 52.84 | 43.92 | 54.07 | 25.35 | 31.03 | 35.04 |
| | - | 1,8 | 51.93 | 43.87 | 54.00 | 25.67 | 31.03 | 35.08 |
| | 8 | 1 | 54.66 | 44.47 | 57.12 | 25.41 | 31.05 | 34.85 |
| | 8 | 2 | 54.33 | 44.10 | 55.75 | 25.82 | 31.05 | 34.80 |
| | 8 | 4 | 54.03 | 43.73 | 54.72 | 26.06 | 31.03 | 34.83 |
| | 8 | 1,2,4 | 54.80 | 44.26 | 56.30 | 26.00 | 31.05 | 34.78 |
| | 4 | - | 82.76 | 62.77 | 77.57 | 33.01 | 31.75 | 38.54 |
| | - | 1,4 | 81.38 | 62.73 | 77.19 | 33.65 | 31.69 | 38.65 |
| | 4 | 1 | 83.47 | 63.00 | 77.92 | 34.18 | 31.80 | 38.66 |
| | 4 | 2 | 83.14 | 63.09 | 77.87 | 34.36 | 31.81 | 38.63 |
| SD3 | 4 | 8 | 83.30 | 62.94 | 78.02 | 34.37 | 31.80 | 38.66 |
| MM-DiT | 8 | - | 82.52 | 62.94 | 77.55 | 33.08 | 31.74 | 38.54 |
| | - | 1,8 | 82.84 | 62.60 | 76.11 | 34.21 | 31.75 | 38.67 |
| | 8 | 1 | 83.45 | 63.16 | 78.18 | 34.50 | 31.81 | 38.71 |
| | 8 | 2 | 82.89 | 62.90 | 77.58 | 34.30 | 31.80 | 38.68 |
| | 8 | 4 | 82.78 | 62.99 | 77.71 | 34.15 | 31.79 | 38.68 |
| | 8 | 1,2,4 | 83.02 | 62.97 | 77.83 | 34.12 | 31.79 | 38.60 |

Table 7: **TSM 2-Stage Ablation on T2I-CompBench.** $n_{core}$ and $n_{context}$ refer to the number of core experts and context experts respectively. Values in green represent the improved performance compared to the 1-stage model with the same core experts, while gray indicate the decreased. The results show that the design of asymmetric TimeStep LoRA experts assembly is better than the symmetric case or without assembly, and $n_1{=}8, n_2{=}1$ can achieve stable performance improvement.

| Metric | FT Method | Value |
|---|---|---|
| | Vanilla LoRA | 14.58 |
| FID↓ | TSM 1-stage | 9.92 ↓4.66 |
| | TSM 2-stage | 9.90 ↓4.68 |
| Patch | Vanilla LoRA | 15.43 |
| -FID↓ | TSM 1-stage | 11.88 ↓3.55 |
| | TSM 2-stage | 11.82 ↓3.61 |
| CLIP- | Vanilla LoRA | 0.3176 |
| Score↑ | TSM 1-stage | 0.3208 ↑%1.01 |
| | TSM 2-stage | 0.3212 ↑%1.13 |

| Model | $z_t$ | $t$ | Color↑ | Shape↑ | Texture↑ | Spatial↑ | Non-Spatial↑ | Complex↑ |
|---|---|---|---|---|---|---|---|---|
| SD1.5 | ✓ | ✓ | **57.12** | **46.65** | **58.16** | **18.80** | **31.83** | 32.94 |
| UNet | ✗ | ✓ | 51.42 | 44.09 | 53.46 | 13.56 | 31.75 | **33.45** |
| | ✓ | ✗ | 53.64 | 46.24 | 55.08 | 16.31 | 31.72 | 33.42 |
| PixArt-$\alpha$ | ✓ | ✓ | **54.66** | 44.47 | **57.12** | **25.41** | **31.05** | 34.85 |
| DiT | ✗ | ✓ | 45.37 | 42.49 | 52.09 | 24.84 | 30.99 | 34.83 |
| | ✓ | ✗ | 47.23 | **44.69** | 54.30 | 25.25 | 30.99 | **34.86** |
| SD3 | ✓ | ✓ | **83.45** | **63.16** | **78.18** | **34.50** | **31.81** | 38.71 |
| MM-DiT | ✗ | ✓ | 80.99 | 60.38 | 74.62 | 31.87 | 31.61 | 38.53 |
| | ✓ | ✗ | 82.55 | 62.25 | 76.68 | 31.53 | 31.74 | **38.87** |

Table 8: **Model Distillation Ablation based on DMD2.**  Table 9: **Gating Ablation on T2I-CompBench.** The model performance is optimal when the router's input has both $z_t$ and $t$.

and training settings are the same as PixArt-$\alpha$ in Sec. 4.2. The settings of PixArt-$\alpha$ and SD3 are in Sec. 4.2 and 4.1. Notably, when $n{=}1$, TSM 1-stage degenerates to vanilla LoRA. As shown in Tab. 6, regardless of whether we train each LoRA for the same steps, introduce equivalent training costs ($n{\times}step{=}32K$) or the same amount of additional parameters, all $n{=}2, 4, 8$ configurations significantly outperform vanilla LoRA. This highlights that the TSM 1-stage surpasses vanilla LoRA. Moreover, we can find that the performance of $n{=}4$ and $n{=}8$ is similar. Therefore, we believe that $n{=}8$ is enough for the division of the overall timesteps.

**Assembling Stage.** We conduct TSM 2-stage ablation experiments on T2I-CompBench, based on TSM 1-stage model with $r{=}4$. The training settings are same as Fostering Stage ablation. As shown in Tab. 7, we ablate the core expert and context expert. It shows that TSM 2-stage can improve model performance in most cases compared to TSM 1-stage. But surprisingly, the number of context LoRA and the performance in 1-stage have little impact on the performance in 2-stage. This is why we use the simplest $n_2{=}1$ of context LoRA in the experimental settings in Sec. 4.1, 4.2, 4.3. We also study on the symmetry of the TimeStep experts without core LoRA in Tab. 7, all the TimeStep experts are context LoRA. The experiment results show that the 2-stage performance of the symmetrical pattern is often worse than the asymmetrical pattern. Finally, as shown in Tab. 9,

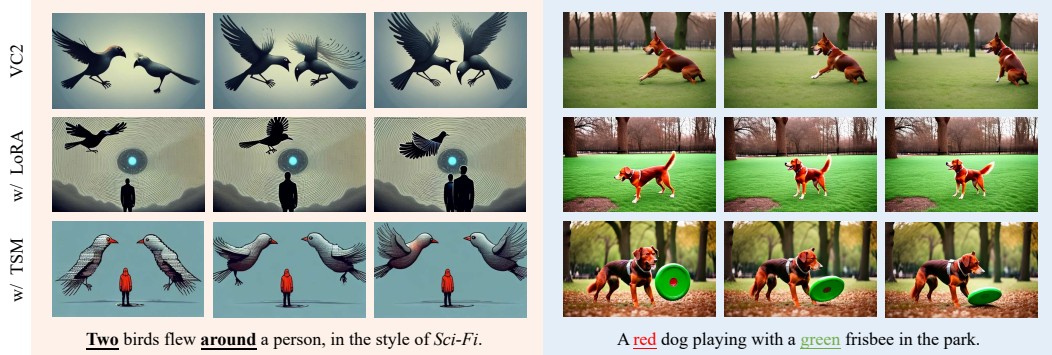

Figure 4: **Comparison on Video Modality.** The videos generated by the LoRA-tuned model are not aligned with the prompts, while our TSM facilitates high-quality and consistent video generation.

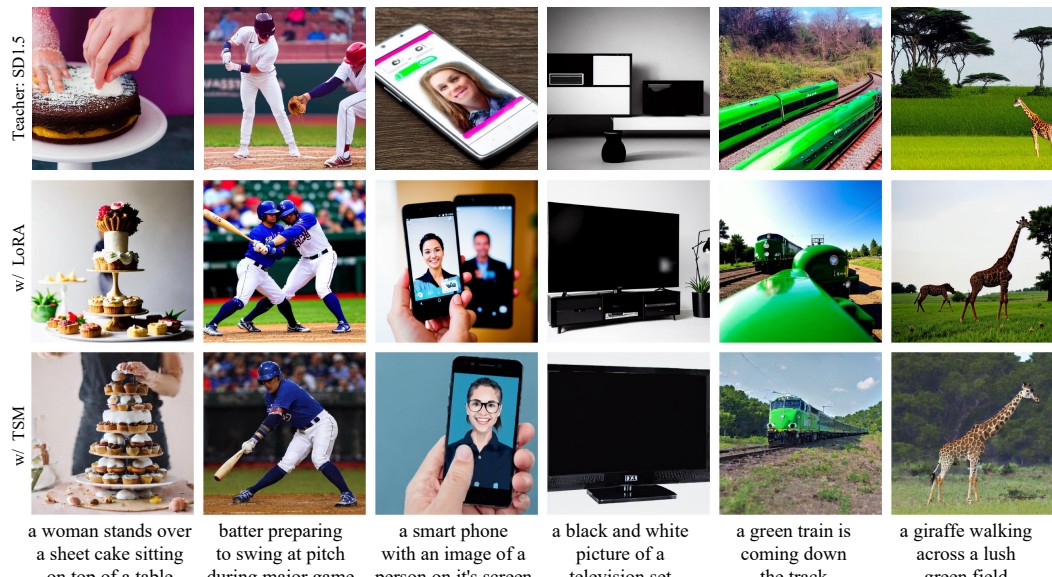

Figure 5: **Comparison on Model Distillation.** The images generated by our TSM better align with the prompts, outperforming the vanilla LoRA, and even surpassing the teacher SD1.5 in some cases.

we conduct ablation experiments on the router's input, and the results show that it is necessary for the router to receive both feature $z_t$ and timestep $t$ as inputs.

**Visualization.** As shown in Fig. 1, in the domain adaptation task, the TSM fine-tuned model revises the incorrect images generated by the pre-trained model, while LoRA could not. As shown in Fig. 1 and 4, in the post-pretraining task, the TSM fine-tuned model improves the alignment between images/videos and text without degrading visual quality, while the LoRA fine-tuned model exhibits a significant decline in both visual quality and vision-text alignment. As shown in Fig. 5, in model distillation task, the TSM fine-tuned model is more aligned with the prompts, outperforming LoRA.

## 5 CONCLUSION

We introduce the TimeStep Master (TSM) paradigm to enhance the fine-tuning of diffusion models. Unlike previous approaches that use a single LoRA for all timesteps, TSM employs different LoRAs on different timestep intervals. Through the fostering and assembling stages, TSM effectively learns diverse noise levels via an asymmetrical mixture of TimeStep LoRA experts. Extensive experiments show that TSM outperforms existing approaches in domain adaptation, post-pretraining, and model distillation. Overall, TSM demonstrates strong generalization across various model architectures and visual modalities, marking a significant advancement in efficient diffusion model tuning.

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
