# OpenReview forum: "TimeStep Master: Asymmetrical Mixture of Timestep LoRA Experts for Versatile and Efficient Diffusion Models in Vision"
_ICLR.cc/2025/Conference — Submitted to ICLR 2025_

### Official Review · Reviewer_E3zt · 2024-10-27

**Soundness:** 2
**Presentation:** 3
**Contribution:** 2
**Rating:** 5
**Confidence:** 5

**Summary:**

This paper, focusing on efficiently fine-tuning diffusion models for vision tasks. proposes a new framework, TSM, which improves the LoRA tuning of diffusion models by using different LoRA modules for various timesteps during training. The finetuning consists of a fostering stage, where TimeStep LoRA experts are trained at different timestep intervals to capture varying noise levels, and an assembling stage for an asymmetrical mixture of TimeStep LoRA experts combining these experts at multiple scales, The authors demonstrate the effectiveness of TSM on several tasks and show TSM outperforms vanilla LoRA methods and achieves state-of-the-art results in multiple tasks.

**Strengths:**

1. Rather than applying the vanilla LoRA, TSM  introduces an innovative way to address the limitations of LoRA in diffusion models by creating specialized experts for different timesteps.
2. TSM achieves SOTA results across multiple tasks on image and video modalities. The results show consistent improvements in model performance and generalization.

**Weaknesses:**

1. The two-stage TSM approach relies on a relatively complex design, which requires heavy fine-tuning and hyperparameter optimization, such as the interval selection and router design. This may not generalize easily to tasks outside the evaluated benchmarks.
2. Extra parameters limit the scalability of the model. It also inherits the inference procedure of the diffusion process, which is also computationally burdensome.
3. Compared to Vanilla LoRA, some improvements, such as in color and shape metrics shown in Table 1, are relatively modest.

**Questions:**

1. The ablation study shows that the increase of r does not necessarily increase the performance. Were other values of r tested? How would the performance change w.r.t the dimension r?
2. Can this framework be applied to more tasks? For example, image restoration or image segmentation. The reviewer wants to see some simple experiments about this.

---

> ### Author Response · Authors · 2024-11-20
> **Reply to Reviewer E3zt (part1)**
>
> Thank you for your valuable reviews, we will answer your questions one by one regarding these weaknesses.
>
> 1. Question: The two-stage TSM approach relies on a relatively complex design and may not generalize easily to tasks outside the evaluated benchmarks.
>
> We have verified the effectiveness of TSM on three classic architectures (U-Net, DiT, MMDiT), two modalities (image, video), and three tasks (domain adaptation, post-pretraining, model distillation). We think the generalization ability and versatility of our TSM have been fully verified. Furthermore, most of our hyperparameter selections are directly followed by the baseline [1, 2]. For additional hyperparameters regarding the selection of core LoRA and context LoRA, there is actually the simplest selection method, which is our We have obtained SOTA settings in multiple experiments, that is, n=8 for core LoRA and n=1 for context LoRA. In addition, no additional transcendent parameters are introduced, so it will be easy to transfer our method to other fields.
>
> [1] Kaiyi Huang, Kaiyue Sun, Enze Xie, Zhenguo Li, and Xihui Liu. T2i-compbench: A comprehensive benchmark for open-world compositional text-to-image generation. Advances in Neural Information Processing Systems, 36:78723–78747, 2023.
>
> [2] Tianwei Yin, Michael Gharbi, Taesung Park, Richard Zhang, Eli Shechtman, Fredo Durand, and William T Freeman. Improved distribution matching distillation for fast image synthesis. arXiv preprint arXiv:2405.14867, 2024a.
>
> 2. Question: Extra parameters limit the scalability of the model. It also inherits the inference procedure of the diffusion process, which is also computationally burdensome.
>
> Because the amount of LoRA parameters we introduced is very small, as shown in the table below, the increased time and memory consumption of our method compared to the origin model is indeed acceptable. Therefore, TSM does not actually affect model scalability.
>
> | Model    | Method       | Diffusion Network               | Text Encoder              | Train Percent (%) |
> |----------|--------------|---------------------------------|---------------------------|-------------------|
> | SD1.5    | Vanilla LoRA | Wq, Wk, Wv, Wout (0.7972M)      | Wq, Wv (0.1475M)          | 0.09604           |
> |          | TSM 1-stage  | Wq, Wk, Wv, Wout (0.7972M)      | Wq, Wv (0.1475M)          | 0.09604           |
> |          | TSM 2-stage  | Rq, Rk, Rv, Rout (0.2275M)      | Wq, Wv (0.04243M)         | 0.02722           |
> | PixArt-α | Vanilla LoRA | Wq, Wk, Wv, Wout (0.2064M)      | Wq, Wv (0.1573M)          | 0.06764           |
> |          | TSM 1-stage  | Wq, Wk, Wv, Wout (0.2064M)      | Wq, Wv (0.1573M)          | 0.06764           |
> |          | TSM 2-stage  | Rq, Rk, Rv, Rout (0.0482M)      | Wq, Wv (0.02446M)         | 0.01344           |
> | SD3      | Vanilla LoRA | Wq, Wk, Wv, Wout (1.18M)        | Wq, Wk, Wv, Wout (1.606M) | 0.05635           |
> |          | TSM 1-stage  | Wq, Wk, Wv, Wout (1.18M)        | Wq, Wk, Wv, Wout (1.607M) | 0.05635           |
> |          | TSM 2-stage  | Rq, Rk, Rv, Rout (0.2435M)      | Wq, Wv (0.3767M)          | 0.01311           |
>
> We have supplemented the following table with a comparison of the time and memory usage required to run the original model and our method on a single A100, where the batch size is set to 1.
>
> | Method           | Time (s) | Memory (GB) |
> |------------------|----------|-------------|
> | SD1.5            | 8.35     | 2.04        |
> | SD1.5+TSM (ours) | 8.52     | 2.04        |
>
> 3. Question: Compared to Vanilla LoRA, some improvements, such as in color and shape metrics shown in Table 1, are relatively modest.
>
> In Table 1, compared to vanilla LoRA, TSM demonstrates notable improvements across all three fundamental architectures (UNet, DiT, and MMDiT). However, the enhancement brought by TSM to SD3 is relatively less pronounced than those observed in SD1.5 and PixArt-α. This is primarily because SD3's performance on T2I-Compbench has already approached saturation levels. Therefore, the fact that TSM achieves approximately 1% improvement in attributes such as color and shape is actually quite remarkable and commendable. Furthermore, as shown in Tables 2 and 3, the advantages of TSM over vanilla LoRA are demonstrated even more comprehensively.

---

> ### Author Response · Authors · 2024-11-20
> **Reply to Reviewer E3zt (part2)**
>
> 4. Question: The ablation study shows that the increase of r does not necessarily increase performance.
>
> All experiments presented in our paper were conducted under the constraint of maintaining equal training costs. When we remove this constraint and train each timestep expert with n=8 and 4000 training steps, the results are shown in the table below. The experimental results demonstrate that the model's performance indeed increases with the growth of r.
>
> | Rank | Color | Shape | Texture | Spatial | Non-spatial | Complex |
> |------|-------|-------|---------|---------|-------------|---------|
> | SD1.5 | 36.97 | 36.27 | 41.25   | 11.04   | 31.05       | 30.79   |
> | 1    | 54.63 | 44.66 | 55.35   | 13.23   | 31.66       | 31.84   |
> | 4    | 56.48 | 45.91 | 57.08   | **18.01**   | 31.77       | 32.79   |
> | 16   | 57.86 | 45.99 | 58.13   | 14.11   | 31.82       | 33.20   |
> | 64   | **59.37** | **46.67** | **58.99** | 15.40   | **31.86** | **33.59** |
>
> 5. Question: Can this framework be applied to more tasks?
>
> We have verified the effectiveness of TSM on three classic architectures (U-Net, DiT, MMDiT), two modalities (image, video), and three tasks (domain adaptation, post-pretraining, model distillation). Due to time constraints, we will complete the comparative experiments of TSM on more downstream tasks in future versions of the paper, such as image and video restoration, image and video segmentation, etc.

---

> > ### Comment · Reviewer_E3zt · 2024-11-23
> > **Response by the reviewer**
> >
> > Thanks for the reply. After comprehensively considering the rebuttal and the comments from other reviewers, I think this paper's current version is below the ICLR acceptance bar. Hence I keep the original score.

---

> > > ### Author Response · Authors · 2024-11-24
> > > **Reply to Reviewer E3zt**
> > >
> > > We understand that you have maintained your original score, and we fully respect your decision.
> > >
> > > However, we sincerely hope that you can approach this novel optimization direction for diffusion models with an open mind and a degree of receptiveness, giving this validated algorithm a chance to grow and develop. We believe that TSM proposes a completely new direction for optimizing diffusion models, specifically focusing on timestep optimization. To support this, we conducted what we believe to be the most representative experiments, including ablation studies and horizontal comparisons. The results consistently demonstrate the effectiveness and necessity of this optimization direction.

---

### Official Review · Reviewer_msCv · 2024-10-30

**Soundness:** 3
**Presentation:** 3
**Contribution:** 2
**Rating:** 5
**Confidence:** 4

**Summary:**

The paper introduces the TimeStep Master (TSM) paradigm for efficiently fine-tuning diffusion models using Low-Rank Adaptation (LoRA). TSM addresses the limitation of applying the same LoRA across all timesteps by introducing two stages: fostering, where different LoRAs are applied to specific timestep intervals, and assembling, which combines these experts asymmetrically with a core-context collaboration. TSM improves diffusion models' ability to handle different noise levels, resulting in better performance across tasks like domain adaptation, post-pretraining, and model distillation, with reduced computational costs.

**Strengths:**

The TimeStep Master (TSM) paradigm addresses the problem of text adherence deterioration in LoRA-tuned diffusion models by introducing different LoRA modules for distinct timesteps, allowing the model to handle varying noise levels more effectively. The use of an asymmetrical mixture of experts, with a core-context collaboration mechanism, helps improve model adaptability to different noise distributions. TSM demonstrates versatility across tasks such as domain adaptation, post-pretraining, and model distillation, achieving strong results on various benchmarks for both images and videos. Its design also maintains relatively low computational costs, making it an efficient option for fine-tuning large models, with results showing broad applicability across different architectures and data modalities.

**Weaknesses:**

1. The paper does not clearly compare its method with non-LoRA approaches that address prompt misalignment issues such as attribute disentanglement, spatial relationships (Fig. 1), generation omissions, and distortions (Fig. 4). These include training-free methods like *Training-Free Structured Diffusion Guidance*, *BoxDiff*, and *Training-Free Layout Control with Cross-Attention Guidance*, or training-required methods like *LayoutDiffusion*. A comparison of computational overhead and training parameters would provide more insight into the advantages of the proposed method.

2. Prompt misalignment issues, such as incorrect spatial relationships and attribute binding errors, are common even in pre-trained or fully fine-tuned models. These problems are not exclusive to LoRA-tuned models and may not be solely caused by the different noise levels at various timesteps, as suggested by the authors. More explanation and insight are needed on how the proposed LoRA tuning method resolves these issues. Additionally, to validate the authors' claim that deterioration in LoRA-tuned models is due to using a single LoRA across all timesteps, it might be sufficient to fine-tune only the UNet parameters without modifying the text encoder. Since the text encoder statically computes embeddings and doesn't adjust based on timestep noise levels, any improvements from tuning it may come from upgrading the text encoder itself. More experiments are necessary to confirm this.

3. Combining multiple LoRAs can sometimes impair a pre-trained model's inherent generative ability or cause individual LoRA characteristics to be lost. It’s unclear whether the gating-based combination method explored the potential degradation in model performance. Including comparisons with other LoRA combination methods, such as *Linear Arithmetic Composition* or *Reference Tuning-Based Composition*, could provide more insights. Moreover, ablation studies on LoRA combination methods (beyond just the gating inputs) would be useful. A related work is *Mixture of LoRA Experts*.

4. As mentioned in point 1, non-LoRA methods can address similar issues, potentially with lower computational cost. Some training-free approaches also achieve good results. It remains unclear if the TSM LoRA-tuned method is necessary or optimal for specific application scenarios. While the method is interesting and simple, its practical use in certain scenarios remains to be seen.

5. In Fig. 4, the "in the style of Sci-Fi" prompt does not seem to be well followed, and in Fig. 5, some images (e.g., the black-and-white image) show poor understanding of the prompt.

**Questions:**

As I understand it, compared to traditional LoRA, the proposed method combines multiple different interval LoRAs at the same timestep, with different LoRAs for different timesteps. Does this only result in an additional 1M parameters? Could the authors clarify more specifically where the 1M increase in training parameters comes from? Is it solely due to the added gating mechanism?

---

> ### Author Response · Authors · 2024-11-20
> **Reply to Reviewer msCv (part1)**
>
> Thank you for your valuable reviews, we will answer your questions one by one regarding these weaknesses.
>
> 1. Question: The paper does not clearly compare its method with non-LoRA approaches that address prompt misalignment issues.
>
> Thanks for the reminder. These methods perform additional training-free operations on the attention module, but their improvement in the essential capabilities of the model is limited. We conducted experiments on SD2 and compared the results with other methods (including training-free methods) as follows. The results demonstrate that TSM still has a large performance advantage compared to training-free methods.
>
> | Method                          | Color | Shape | Texture | Spatial | Non-spatial | Complex |
> |---------------------------------|-------|-------|---------|---------|-------------|---------|
> | SD2 [1]      | 50.65 | 42.21 | 49.22   | 13.42   | 31.27       | 33.86   |
> | SD2 + Composable [2] | 40.63 | 32.99 | 36.45   | 8.00    | 29.80       | 28.98   |
> | SD2 + Structured [3]  | 49.90 | 42.18 | 49.00   | 13.86   | 31.11       | 33.55   |
> | SD2 + Attn Exct [4] | 64.00 | 45.17 | 59.63   | 14.55   | 31.09       | 34.01   |
> | SD2 + GORS unbiased [5] | 64.14 | 45.46 | 60.25   | 17.25   | 31.58       | 34.70   |
> | SD2 + GORS [5]      | 66.03 | 47.85 | 62.87   | 18.15   | **31.93**       | 33.28   |
> | SD2 + TSM (Ours)       | **75.93** | **54.34** | **67.44**   | **18.34**   | 31.47       | **34.20**   |
>
> [1] Robin Rombach, Andreas Blattmann, Dominik Lorenz, Patrick Esser, and Bj¨orn Ommer. Highresolution image synthesis with latent diffusion models. In Proceedings of the IEEE/CVF conference on computer vision and pattern recognition, pp. 10684–10695, 2022a.
>
> [2] N. Liu, S. Li, Y. Du, A. Torralba, and J. B. Tenenbaum, “Compositional visual generation with composable diffusion models,” in ECCV, 2022.
>
> [3] W. Feng, X. He, T.-J. Fu, V. Jampani, A. Akula, P. Narayana, S. Basu, X. E. Wang, and W. Y. Wang, “Training-free structured diffusion guidance for compositional text-to-image synthesis,” in ICLR, 2023.
>
> [4] H. Chefer, Y. Alaluf, Y. Vinker, L. Wolf, and D. Cohen-Or, “Attend-andexcite: Attention-based semantic guidance for text-to-image diffusion models,” in ACM Trans. Graph., 2023.
>
> [5] Kaiyi Huang, Kaiyue Sun, Enze Xie, Zhenguo Li, and Xihui Liu. T2i-compbench: A comprehensive benchmark for open-world compositional text-to-image generation. Advances in Neural Information Processing Systems, 36:78723–78747, 2023.
>
> 2. Question: A comparison of computational overhead and training parameters would provide more insight into the advantages of the proposed method.
>
> Because the amount of LoRA parameters we introduced is very small, as shown in the table below, the increased time and memory consumption of our method compared to the origin model is acceptable.
>
> | Model    | Method       | Diffusion Network               | Text Encoder              | Train Percent (%) |
> |----------|--------------|---------------------------------|---------------------------|-------------------|
> | SD1.5    | Vanilla LoRA | Wq, Wk, Wv, Wout (0.7972M)      | Wq, Wv (0.1475M)          | 0.09604           |
> |          | TSM 1-stage  | Wq, Wk, Wv, Wout (0.7972M)      | Wq, Wv (0.1475M)          | 0.09604           |
> |          | TSM 2-stage  | Rq, Rk, Rv, Rout (0.2275M)      | Wq, Wv (0.04243M)         | 0.02722           |
> | PixArt-α | Vanilla LoRA | Wq, Wk, Wv, Wout (0.2064M)      | Wq, Wv (0.1573M)          | 0.06764           |
> |          | TSM 1-stage  | Wq, Wk, Wv, Wout (0.2064M)      | Wq, Wv (0.1573M)          | 0.06764           |
> |          | TSM 2-stage  | Rq, Rk, Rv, Rout (0.0482M)      | Wq, Wv (0.02446M)         | 0.01344           |
> | SD3      | Vanilla LoRA | Wq, Wk, Wv, Wout (1.18M)        | Wq, Wk, Wv, Wout (1.606M) | 0.05635           |
> |          | TSM 1-stage  | Wq, Wk, Wv, Wout (1.18M)        | Wq, Wk, Wv, Wout (1.607M) | 0.05635           |
> |          | TSM 2-stage  | Rq, Rk, Rv, Rout (0.2435M)      | Wq, Wv (0.3767M)          | 0.01311           |
>
> We have supplemented the following table with a comparison of the time and memory usage required to run the original model and our method on a single A100, where the batch size is set to 1.
>
> | Method           | Time (s) | Memory (GB) |
> |------------------|----------|-------------|
> | SD1.5            | 8.35     | 2.04        |
> | SD1.5+TSM (ours) | 8.52     | 2.04        |

---

> ### Author Response · Authors · 2024-11-20
> **Reply to Reviewer msCv (part2)**
>
> 3. Question: The prompt misalignment problem is not exclusive to LoRA-tuned models and may not be solely caused by the different noise levels at various timesteps.
>
> Thank you for the analysis. Perhaps there are various reasons for the degradation of model capabilities caused by LoRA, such as the low-rank characteristics of the matrix, insufficient training, etc. However, TSM can largely solve the degradation problem caused by LoRA under the premise of approximate training overhead. Therefore, we believe that TSM will replace LoRA as the preferred solution for low-cost fine-tuning of diffusion models.
>
> 4. Question: It might be sufficient to fine-tune only the UNet parameters without modifying the text encoder.
>
> As shown in Table 1, 2, 3, 4, 5, 6, our comparison experiments between vanilla LoRA and TSM, the modules for adding LoRA are all the same, where LoRA is consistently added to the text encoder. Therefore, our ablation experiments are fair and can fully prove that our TSM is significantly better than vanilla LoRA. Besides, most of our hyperparameter selections (including whether to train a text encoder or not) are directly followed by the baselines [1, 2].
>
> [1] Kaiyi Huang, Kaiyue Sun, Enze Xie, Zhenguo Li, and Xihui Liu. T2i-compbench: A comprehensive benchmark for open-world compositional text-to-image generation. Advances in Neural Information Processing Systems, 36:78723–78747, 2023.
>
> [2] Tianwei Yin, Michael Gharbi, Taesung Park, Richard Zhang, Eli Shechtman, Fredo Durand, and
> William T Freeman. Improved distribution matching distillation for fast image synthesis. arXiv preprint arXiv:2405.14867, 2024a.
>
> 5. Question: It’s unclear whether the gating-based combination method explored the potential degradation in model performance.
>
> Table 9 in our article shows our ablation experiments on router architecture. Besides, the experimental results in Tables 1, 2, and 3 all show that even such a simple router architecture is enough to make our method better than existing methods.
>
> 6. Question: Ablation studies on LoRA combination methods (beyond just the gating inputs) would be useful. A related work is a Mixture of LoRA Experts.
>
> In the table below, we further add a comparison of MoE LoRA trained from scratch (even though we are working at the same time) with TSM. It can be seen that our TSM still achieves the best results.
>
> | Method            | Color | Shape  | Texture | Spatial | Non-spatial | Complex |
> |-------------------|-------|--------|---------|---------|-------------|---------|
> | SD1.5             | 36.97 | 36.27  | 41.25   | 11.04   | 31.05       | 30.79   |
> | SD+MoE LoRA       | 52.26 | 39.83  | 51.38   | 11.71   | 31.34       | 32.00   |
> | SD1.5+TSM 1-stage | 56.48 | 45.91  | 57.08   | 18.01   | 31.77       | **32.79** |
> | SD1.5+TSM 2-stage | **57.59** | **46.18** | **57.69** | **17.91** | **31.82** | 32.78   |
>
> 7. Question: Its practical use in certain scenarios remains to be seen.
>
> We have verified the effectiveness of TSM on three classic architectures (U-Net, DiT, MMDiT), two modalities (image, video), and three tasks (domain adaptation, post-pretraining, model distillation). We think the practicality and versatility of our TSM have been fully verified.
>
> 8. Question: In Fig. 4, the "in the style of Sci-Fi" prompt does not seem to be well followed, and in Fig. 5, some images (e.g., the black-and-white image) show poor understanding of the prompt.
>
> The Sci-Fi  actually denotes Science-Fiction, and the test prompts are selected from EvalCrafter benchmark. To be noted, in the example shown on the left side of Fig. 4, the result generated by VC2 lacks a crucial main character, while the result from VC2 with LoRA is missing a bird. The absence of the main subject represents a relatively more significant error. In the example presented in Fig. 5 (the black-and-white image), SD1.5 has generated numerous peculiar elements, and the image produced by SD1.5 with LoRA even includes green vegetation, which evidently deviates substantially from the input prompt. Compared to other methods, our method has greatly improved the problem of mismatch between text and generated contents. Although there may still be some small mismatches, it does improve a lot.

---

> ### Author Response · Authors · 2024-11-20
> **Reply to Reviewer msCv (part3)**
>
> 9. Question: Does this only result in an additional 1M parameters? Could the authors clarify more specifically where the 1M increase in training parameters comes from? Is it solely due to the added gating mechanism?
>
> Thank you for the suggestion. We added the analysis of parameters. As shown in the table below, because the structure of the router is very simple, the additional training parameters introduced in the second stage are only 1M and the increased parameters of TSM compared to the origin model are acceptable.
>
> | Model    | Method       | Diffusion Network               | Text Encoder              | Train Percent (%) |
> |----------|--------------|---------------------------------|---------------------------|-------------------|
> | SD1.5    | Vanilla LoRA | Wq, Wk, Wv, Wout (0.7972M)      | Wq, Wv (0.1475M)          | 0.09604           |
> |          | TSM 1-stage  | Wq, Wk, Wv, Wout (0.7972M)      | Wq, Wv (0.1475M)          | 0.09604           |
> |          | TSM 2-stage  | Rq, Rk, Rv, Rout (0.2275M)      | Wq, Wv (0.04243M)         | 0.02722           |
> | PixArt-α | Vanilla LoRA | Wq, Wk, Wv, Wout (0.2064M)      | Wq, Wv (0.1573M)          | 0.06764           |
> |          | TSM 1-stage  | Wq, Wk, Wv, Wout (0.2064M)      | Wq, Wv (0.1573M)          | 0.06764           |
> |          | TSM 2-stage  | Rq, Rk, Rv, Rout (0.0482M)      | Wq, Wv (0.02446M)         | 0.01344           |
> | SD3      | Vanilla LoRA | Wq, Wk, Wv, Wout (1.18M)        | Wq, Wk, Wv, Wout (1.606M) | 0.05635           |
> |          | TSM 1-stage  | Wq, Wk, Wv, Wout (1.18M)        | Wq, Wk, Wv, Wout (1.607M) | 0.05635           |
> |          | TSM 2-stage  | Rq, Rk, Rv, Rout (0.2435M)      | Wq, Wv (0.3767M)          | 0.01311           |

---

### Official Review · Reviewer_iqHj · 2024-11-02

**Soundness:** 3
**Presentation:** 3
**Contribution:** 3
**Rating:** 6
**Confidence:** 4

**Summary:**

The paper finds that using the same LoRA for fine-tuning diffusion models at different timesteps has its limitations. To address this, this paper proposes the TSM paradigm, which employs different LoRA adaptations at various timesteps and integrates them through a novel asymmetrical mixture, achieving state-of-the-art performance across multiple tasks and model structures.

**Strengths:**

1. This paper identifies a key limitation in using a single LoRA for fine-tuning diffusion models: the significant differences in feature distributions across different timesteps make it challenging for a single LoRA to effectively learn all information. To address the above issue, the paper introduces a two-stage approach that leverages multiple LoRA modules and an ensemble method.
2. Comprehensive experiments are conducted across several tasks, such as domain adaptation, post-pretraining, and model distillation, all yielding promising results.
3. The paper’s illustrations clearly depict the details of the proposed method.

**Weaknesses:**

1. Since TSM uses a router to dynamically assemble experts based on timestep and previous step results, each timestep may involve multiple expert calls. This results in higher memory usage and greater latency during inference.
2. Although the assembling stage can be understood from the flowchart, the explanation of Equation (7) may be somewhat confusing, especially regarding the subscript notation of $\epsilon$, which lacks clarity.
3. Regarding the results in Table 1, the paper presents various outcomes for SD2 and SD2+X, but the results for SD2 with the proposed method are missing. What could be the reason for this?
4. In Table 2, the authors provide results on T2I-CompBench and EvalCrafter, yet FID is also a widely accepted and important metric in pretraining works such as Pixart-alpha. Could an evaluation based on this metric be added?
5. Table 6 shows the ablation study for the first stage of TSM, where it’s observed that the overall performance improves with 8 experts. Would further increasing the number of experts theoretically enhance the results even more?

**Questions:**

Please refer to the weaknesses part.

---

> ### Author Response · Authors · 2024-11-20
> **Reply to Reviewer iqHj (part1)**
>
> Thank you for your valuable reviews, we will answer your questions one by one regarding these weaknesses.
>
> 1. Question: This results in higher memory usage and greater latency during inference.
>
> Thank you for your concern. We have added the analysis of the inference memory and time usage. For training, because the amount of LoRA parameters we introduced is very small, as shown in the table below, the increased time and memory consumption of our method compared to the origin model is acceptable.
>
> | Model    | Method       | Diffusion Network               | Text Encoder              | Train Percent (%) |
> |----------|--------------|---------------------------------|---------------------------|-------------------|
> | SD1.5    | Vanilla LoRA | Wq, Wk, Wv, Wout (0.7972M)      | Wq, Wv (0.1475M)          | 0.09604           |
> |          | TSM 1-stage  | Wq, Wk, Wv, Wout (0.7972M)      | Wq, Wv (0.1475M)          | 0.09604           |
> |          | TSM 2-stage  | Rq, Rk, Rv, Rout (0.2275M)      | Wq, Wv (0.04243M)         | 0.02722           |
> | PixArt-α | Vanilla LoRA | Wq, Wk, Wv, Wout (0.2064M)      | Wq, Wv (0.1573M)          | 0.06764           |
> |          | TSM 1-stage  | Wq, Wk, Wv, Wout (0.2064M)      | Wq, Wv (0.1573M)          | 0.06764           |
> |          | TSM 2-stage  | Rq, Rk, Rv, Rout (0.0482M)      | Wq, Wv (0.02446M)         | 0.01344           |
> | SD3      | Vanilla LoRA | Wq, Wk, Wv, Wout (1.18M)        | Wq, Wk, Wv, Wout (1.606M) | 0.05635           |
> |          | TSM 1-stage  | Wq, Wk, Wv, Wout (1.18M)        | Wq, Wk, Wv, Wout (1.607M) | 0.05635           |
> |          | TSM 2-stage  | Rq, Rk, Rv, Rout (0.2435M)      | Wq, Wv (0.3767M)          | 0.01311           |
>
> For inference, we have supplemented the following table with a comparison of the time and memory usage required to run the original model and our method on a single A100, where the batch size is set to 1.
>
> | Method           | Time (s) | Memory (GB) |
> |------------------|----------|-------------|
> | SD1.5            | 8.35     | 2.04        |
> | SD1.5+TSM (ours) | 8.52     | 2.04        |
>
> 2. Question: Although the assembling stage can be understood from the flowchart, the explanation of Equation (7) may be somewhat confusing, especially regarding the subscript notation of ϵ, which lacks clarity.
>
> Thank you for your detailed reminder. As we stated in sections 139-140 of the article, ϵ is the noise sampled from a Gaussian distribution. We are sorry for the misunderstanding.
>
> 3. Question: The results for SD2 with the proposed method are missing. What could be the reason for this?
>
> In our paper, we chose a UNet-based model (SD1.5), a DiT-based model (PixArt-α) and a MMDiT-based model (SD3) as our pretrained model. The reason we chose SD1.5 instead of SD2 is that SD1.5 seems to be more widely circulated on major platforms. The results show that by using SD1.5+TSM significantly exceeds the SD1.5 and its various variants. To further verify our observation, we added the experimental results of TSM on SD2 in the following table. It can be seen that we achieved the most advanced results on SD2 as well.
>
> | Method                          | Color | Shape | Texture | Spatial | Non-spatial | Complex |
> |---------------------------------|-------|-------|---------|---------|-------------|---------|
> | SD2 [1]      | 50.65 | 42.21 | 49.22   | 13.42   | 31.27       | 33.86   |
> | SD2 + Composable [2] | 40.63 | 32.99 | 36.45   | 8.00    | 29.80       | 28.98   |
> | SD2 + Structured [3]  | 49.90 | 42.18 | 49.00   | 13.86   | 31.11       | 33.55   |
> | SD2 + Attn Exct [4] | 64.00 | 45.17 | 59.63   | 14.55   | 31.09       | 34.01   |
> | SD2 + GORS unbiased [5] | 64.14 | 45.46 | 60.25   | 17.25   | 31.58       | 34.70   |
> | SD2 + GORS [5]      | 66.03 | 47.85 | 62.87   | 18.15   | **31.93**       | 33.28   |
> | SD2 + TSM (Ours)       | **75.93** | **54.34** | **67.44**   | **18.34**   | 31.47       | **34.20**   |
>
> [1] Robin Rombach, Andreas Blattmann, Dominik Lorenz, Patrick Esser, and Bj¨orn Ommer. Highresolution image synthesis with latent diffusion models. In Proceedings of the IEEE/CVF conference on computer vision and pattern recognition, pp. 10684–10695, 2022a.
>
> [2] N. Liu, S. Li, Y. Du, A. Torralba, and J. B. Tenenbaum, “Compositional visual generation with composable diffusion models,” in ECCV, 2022.
>
> [3] W. Feng, X. He, T.-J. Fu, V. Jampani, A. Akula, P. Narayana, S. Basu, X. E. Wang, and W. Y. Wang, “Training-free structured diffusion guidance for compositional text-to-image synthesis,” in ICLR, 2023.
>
> [4] H. Chefer, Y. Alaluf, Y. Vinker, L. Wolf, and D. Cohen-Or, “Attend-andexcite: Attention-based semantic guidance for text-to-image diffusion models,” in ACM Trans. Graph., 2023.
>
> [5] Kaiyi Huang, Kaiyue Sun, Enze Xie, Zhenguo Li, and Xihui Liu. T2i-compbench: A comprehensive benchmark for open-world compositional text-to-image generation. Advances in Neural Information Processing Systems, 36:78723–78747, 2023.

---

> ### Author Response · Authors · 2024-11-20
> **Reply to Reviewer iqHj (part2)**
>
> 4. Question: Could an evaluation based on FID be added?
>
> Thank you for the suggestion. We added the results of our FID experiment on PixArt are shown in the following table. PixArt-α's result is measured by ourselves since the model they published is not the model used to measure FID in their paper.
>
> | Method       | FID   |
> |--------------|-------|
> | PixArt-α     | 27.26 |
> | PixArt-α+TSM | 23.64 |
>
> 5. Question: Would further increasing the number of experts theoretically enhance the results even more?
>
> Intuitively, it seems this way, but our experiments have shown that this is not the case. The premise that must be ensured in our experiments is that with the same training cost, as the timesteps are further subdivided and the number of experts increases, the number of steps each expert is trained on decreases. The experiments in Table 6 indicate that when n≤4, the model's performance improves as the number of experts increases. However, if n continues to increase, the model's performance may even decline. We supplemented the experimental result of n=16 as follows, which further verified our conjecture.
>
> | n    | Color | Shape | Texture | Spatial | Non-spatial | Complex |
> |------|-------|-------|---------|---------|-------------|---------|
> | SD1.5 | 36.97 | 36.27 | 41.25   | 11.04   | 31.05       | 30.79   |
> | 1    | 51.86 | 44.74 | 55.74   | 15.70    | 31.70        | 29.84   |
> | 2    | 54.3  | 45.25 | 57.26   | 17.05   | **31.79**   | 31.50    |
> | 4    | 55.85 | **46.45** | **58.06** | **18.32** | 31.77       | 32.95   |
> | 8    | **56.48** | 45.91 | 57.08   | 18.01   | 31.77       | 32.79   |
> | 16   | 54.20  | 45.61 | 56.39   | 14.56   | 31.71       | **33.16** |

---

> > ### Comment · Reviewer_iqHj · 2024-11-26
> > **response to rebuttal**
> >
> > Thanks for your reply. I keep my scores as I think the method of exploring different LoRA modules for distinct timesteps is new. You can focus on addressing others' opinions to increase the probability of raising the score.

---

### Official Review · Reviewer_uR7t · 2024-11-04

**Soundness:** 2
**Presentation:** 3
**Contribution:** 2
**Rating:** 5
**Confidence:** 4

**Summary:**

The authors propose a novel LoRA-based fine-tuning method to efficiently adapt large diffusion-based image generation models. They identify limitations in LoRA-tuned models, noting that diffusion models process noisy inputs differently at various time steps, which restricts their generative capability.
To address this, the paper introduces the TimeStep Master (TSM) paradigm, which incorporates different LoRA matrices for distinct time steps, enabling the learning of various processing modes. TSM comprises two main stages:
Fostering Stage: This phase segments the training process into timestep intervals, each using different LoRA modules.
Assembling Stage: This phase combines the TimeStep LoRA experts, facilitating core-context collaboration.
TSM achieves state-of-the-art performance in domain adaptation, post-pretraining, and model distillation tasks. Furthermore, it demonstrates robust generalization across diverse model architectures and visual data modalities.

**Strengths:**

The authors effectively address the limitations of a single shared LoRA across the entire diffusion process by partitioning it into multiple intervals, enhancing adaptability.
They recognize the efficiency benefits of shared timestep experts, avoiding the need for individual experts for each timestep and proposing a collaborative learning function to maintain efficiency.
The paper is well-structured and provides compelling qualitative and quantitative results.

**Weaknesses:**

The rationale for using different scales of intervals in the assembling stage is not clearly explained.
The paper lacks a comparison between the asymmetrical MoE (Mixture of Experts) and the standard MoE.

Some other comments:

In Figure 3, if only a single scale with n=8 intervals were used, would the vanilla MoE already be capable of learning how to combine the context experts with the core expert through top-2 gating?

In addition, the LoRA projection and reconstruction aim to find the best rank for attention. It seems that the rank is chosen as a hyperparameter. Is it not clear why such a choice is made if this is the real rank of the target matrix. In addition, it is hard to justify why different blocks will share the same rank.

**Questions:**

See weakness

---

> ### Author Response · Authors · 2024-11-20
> **Reply to Reviewer uR7t (part1)**
>
> Thank you for your valuable reviews, we will answer your questions one by one regarding these weaknesses.
>
> 1. Question：The reason for using different scales of intervals in the assembling stage is not clearly explained.
>
> According to our exploration during the experiment, under different timestep conditions, the variances between the features in the diffusion model are very different, and according to the lines 072-075 of our paper, previous work [1, 2] has also analyzed that there are substantial variance discrepancies between the model's intermediate features under different timestep conditions. Therefore, we believe it would be challenging to fit such widely divergent distributions using identical parameters. This reasoning led us to propose our TSM.
>
> [1] Yogesh Balaji, Seungjun Nah, Xun Huang, Arash Vahdat, Jiaming Song, Qinsheng Zhang, Karsten Kreis, Miika Aittala, Timo Aila, Samuli Laine, et al. ediff-i: Text-to-image diffusion models with  an ensemble of expert denoisers. arXiv preprint arXiv:2211.01324, 2022.
>
> [2] Ma, Xinyin and Fang, Gongfan and Wang, Xinchao. DeepCache: Accelerating Diffusion Models for Free
> . CVPR, 2024.
>
> 2. Question: The paper lacks a comparison between the asymmetrical MoE (Mixture of Experts) and the standard MoE.
>
> In the following table, we add the comparison between MoE LoRA trained from scratch and TSM, where TSM still achieves the best results. Furthermore, in Table 7 of our paper, on the premise of inheriting the well-trained timestep LoRA experts from TSM 1-stage, we compared asymmetric MoE and symmetric MoE under three model architectures. Experiments proved that asymmetric MoE is better than Common MoE.
>
> | Method                           | Color | Shape | Texture | Spatial | Non-spatial | Complex |
> |----------------------------------|-------|-------|---------|---------|-------------|---------|
> | SD1.5                            | 36.97 | 36.27 | 41.25   | 11.04   | 31.05       | 30.79   |
> | SD+MoE LoRA                      | 52.26 | 39.83 | 51.38   | 11.71   | 31.34       | 32.00   |
> | SD1.5+TSM 1-stage                | 56.48 | 45.91 | 57.08   | 18.01   | 31.77       | 32.79   |
> | SD1.5+TSM 2-stage (symmetric)    | 54.56 | 45.52 | 56.30   | 17.90   | 31.78       | **33.27**   |
> | SD1.5+TSM 2-stage                | **57.59** | **46.18** | **57.69**   | **17.91**   | **31.82**       | 32.78   |
>
> 3. Question: In Figure 3, if only a single scale with n=8 intervals were used, would the vanilla MoE already be capable of learning how to combine the context experts with the core expert through top-2 gating?
>
> Thank you for your thoughts on our paper and your suggestions for improvements. In fact, we have done this experiment before, and the results are shown in the table below. Our conclusion is that if LoRA trained under the same interval is used as experts in MoE, the performance of the model cannot continue to improve. We speculate that it is because LoRA trained under the same interval can only specialize in the timestep condition within the training range. For example, LoRA trained under large noise levels does not help LoRA trained under small noise levels. Therefore, we use LoRA trained under different intervals as MoE experts in TSM. A large number of experimental results in the paper also prove that our asymmetric design is effective.
>
> | Method                                                            | Color | Shape | Texture | Spatial | Non-spatial | Complex |
> |-------------------------------------------------------------------|-------|-------|---------|---------|-------------|---------|
> | SD1.5+TSM 1-stage                                                 | 56.48 | 45.91 | 57.08   | 18.01   | 31.77       | 32.79   |
> | SD1.5+MoE (experts initialized from n=8 TSM 1-stage)              | 51.64 | 44.24 | 53.08   | 14.31   | 31.72       | **33.42**   |
> | SD1.5+TSM 2-stage                                                 | **57.59** | **46.18** | **57.69**   | **17.91**   | **31.82**       | 32.78   |
> | PixArt-α+TSM 2-stage                                              | 52.84 | 43.92 | 54.07   | 25.35   | 31.03       | 35.04   |
> | PixArt-α+MoE (experts initialized from n=8 TSM 1-stage)           | 45.23 | 42.69 | 52.30   | 24.75   | 30.99       | **34.86**   |
> | PixArt-α+TSM 1-stage                                              | **54.66** | **44.47** | **57.12**   | **25.41**   | **31.04**       | 34.85   |

---

> ### Author Response · Authors · 2024-11-20
> **Reply to Reviewer uR7t (part2)**
>
> 4. Question: It is not clear why such a choice is made if this is the real rank of the target matrix. In addition, it is hard to justify why different blocks share the same rank.
>
> The rank of LoRA is not the real rank of the parameter matrix. Instead, it is used to fine-tune the parameter matrix by approximating it to a low-rank matrix. Our setting of r directly follows the rank used in the previous work [1, 2], which shows that the generalization of our method is superior. In addition, in the table below, we added the ablation experiment between r and model performance when n=8. It is obvious that, as r increases, the performance of the model also increases significantly. For the sake of simplicity, using the same LoRA in different blocks allows us to quickly start training a new model. If the optimal parameters are searched for each block, the calculation cost will be relatively large.
>
> | Rank | Color | Shape | Texture | Spatial | Non-spatial | Complex |
> |------|-------|-------|---------|---------|-------------|---------|
> | SD1.5 | 36.97 | 36.27 | 41.25   | 11.04   | 31.05       | 30.79   |
> | 1    | 54.63 | 44.66 | 55.35   | 13.23   | 31.66       | 31.84   |
> | 4    | 56.48 | 45.91 | 57.08   | **18.01**   | 31.77       | 32.79   |
> | 16   | 57.86 | 45.99 | 58.13   | 14.11   | 31.82       | 33.20   |
> | 64   | **59.37** | **46.67** | **58.99** | 15.40   | **31.86** | **33.59** |
>
> [1] Kaiyi Huang, Kaiyue Sun, Enze Xie, Zhenguo Li, and Xihui Liu. T2i-compbench: A comprehensive benchmark for open-world compositional text-to-image generation. Advances in Neural Information Processing Systems, 36:78723–78747, 2023.
>
> [2] Tianwei Yin, Michael Gharbi, Taesung Park, Richard Zhang, Eli Shechtman, Fredo Durand, and
> William T Freeman. Improved distribution matching distillation for fast image synthesis. arXiv preprint arXiv:2405.14867, 2024a.

---

> > ### Comment · Reviewer_uR7t · 2024-11-23
> >
> > I first thank the authors for providing the additional information.
> >
> > However, I am personally not convinced by using Mixture of TimeStep LoRA. While this is an interesting idea, I think the performance largely depends on the choice of rank, which cannot be explained. So, it somehow looks like I have more experts with varying ranges, so I have more parameters for processing, and therefore better results.
> >
> > I agree with the authors that LoRA only approximates true low-rank space. In this work, using different approximations or making different assumptions on the rank must lead to controversial information competing from one scale to another. However, this field is not discussed or investigated.
> >
> > Finally, I don't really agree with the authors' statement: "It is obvious that, as r increases, the model's performance also increases significantly". If this is the case, it is again contradictory to the proposed Mixture of TimeStep LoRA Experts with smaller ranks for "fine-grained details".
> >
> > To conclude, I think this paper investigates the right direction. However, there are still many issues that are not clear in the current version. So I tend to keep my original ranking.

---

> > > ### Author Response · Authors · 2024-11-24
> > > **Reply to Reviewer uR7t**
> > >
> > > Thank you for your response. In the following reply, I will try to address the concerns you raised regarding our work as clearly as possible.
> > >
> > > In your response, you mentioned: “it somehow looks like I have more experts with varying ranges, so I have more parameters for processing, and therefore better results.” However, this is not the case. Please refer to the ablation study on TSM 1-Stage in Table. 6 of our paper, the premise of this ablation study is that the training cost remains the same, even if the number of parameters differs. For instance, under the setting of n=1, r=4, LoRA is trained for 32,000 steps, whereas under the setting of n=8, r=4, each LoRA is only trained for 4,000 steps because 8 * 4,000 steps = 32,000 steps. This means that what we have discovered is essentially a better optimization algorithm for diffusion models, specifically optimizing timesteps in segments.
> > >
> > > Your statement that we simply have more parameters to process and therefore achieve better results does not hold because our training cost is identical. To further clarify, let me provide an example: the parameter size of GPT-3 is over 100 times larger than GPT-2. However, if GPT-3 were trained with the same computational cost as GPT-2, it would not receive sufficient training, even though it has significantly more parameters. Research into scaling laws focuses on how to optimize large language models under the same training cost to achieve better performance, rather than merely increasing the number of parameters to make the model stronger. Similarly, we believe that the essence of TSM lies in proposing a better optimization algorithm, one that enables us to train superior models while maintaining the same training cost.
> > >
> > > Regarding the viewpoint you disagreed with, “It is obvious that, as r increases, the model's performance also increases significantly,” I apologize for not making this point clear in my earlier response. What this experiment aims to demonstrate is that, under the same number of training steps, the model's performance continues to improve as r increases. This does not contradict the explanation we provided above. In this particular experiment, the training cost is not equal, because a larger r results in higher memory consumption and longer training times for the same number of training steps.
> > >
> > > Regarding the ablation study on r, we aim to show that the choices of n and r are not contradictory. Fine-grained timestep division can also be combined with larger r values, paired with acceptable increases in computational cost, to further optimize the model's performance.
> > >
> > > Additionally, even though our model's parameter size has increased, the inference time and memory consumption have not increased significantly, as shown in the table below:
> > >
> > > | Method           | Time (s) | Memory (GB) |
> > > |------------------|----------|-------------|
> > > | SD1.5            | 8.35     | 2.04        |
> > > | SD1.5+TSM (ours) | 8.52     | 2.04        |
> > >
> > > We believe that TSM proposes a fine-tuning optimization algorithm that is more suitable for diffusion models. This algorithm achieves better results under a fair comparison of identical computational costs. Furthermore, we conducted extensive experiments to validate the adaptability of this optimization algorithm (across three types of diffusion model architectures, two modalities, and three common tasks). These experiments comprehensively demonstrate the applicability of TSM, which can be seamlessly substituted into any scenario where LoRA is used for diffusion models, without incurring additional costs, thereby improving performance.

---

### Author Response · Authors · 2024-11-20
**To All Reviewers**

We sincerely appreciate the time and effort you have dedicated to reviewing our paper! Your valuable feedbacks have been carefully considered, and we will provide point-to-point responses to your reviews in respective reply. We remain open to any additional feedback you may have. If you feel it is appropriate based on our responses, we would be extremely grateful if you could consider raising our score.

---

### Meta-Review · Area_Chair_8nFJ · 2024-12-19

**Metareview:**

The paper introduces TimeStep Master (TSM), a novel paradigm for fine-tuning diffusion models in vision tasks that uses asymmetrical mixtures of timestep-specific LoRA experts to improve versatility and efficiency, achieving state-of-the-art results across various architectures and data modalities.

The reviewers have expressed concerns regarding the rationale behind the choice of different scales of intervals, the comparison with standard MoE, increased memory usage and latency during inference, the need for more detailed explanations, further comparisons with non-LoRA approaches, along with relatively modest improvements in some metrics compared to Vanilla LoRA.
Despite the authors' efforts in the rebuttal, some concerns remain unresolved. Therefore, the final majority of negative ratings lead to a rejection for this submission.

**Additional Comments On Reviewer Discussion:**

- Reviewer uR7t questions the rationale behind the choice of different scales of intervals and the comparison with standard MoE, despite acknowledging the novelty and effectiveness of TSM in addressing the limitations of a single shared LoRA across the entire diffusion process.
- Reviewer iqHj raises concerns about increased memory usage and latency during inference, along with the need for more detailed explanations of the method, while appreciating the identification of a key limitation in using a single LoRA for fine-tuning diffusion models.
- Reviewer msCv suggests further comparisons with non-LoRA approaches and additional experiments to confirm the method's advantages, recognizing the potential in exploring different LoRA modules for distinct timesteps.
- Reviewer E3zt commends the SOTA results achieved by TSM but expresses reservations about the complexity and scalability of the two-stage approach, as well as relatively modest improvements in some metrics compared to Vanilla LoRA.

After rebuttal and discussions, all reviewers keep the rating unchanged.

---

### Decision · Program_Chairs · 2025-01-22

Reject